# Boosting Weakly-Supervised Referring Image Segmentation via Progressive Comprehension

**Zaiquan Yang,  Yuhao Liu**[†]**,  Jiaying Lin,  Gerhard Hancke**[†]**,  Rynson W.H. Lau**[†]

Department of Computer Science
City University of Hong Kong

{zaiquyang2-c, yuhliu9-c,jiayinlin5-c}@my.cityu.edu.hk
{gp.hancke, Rynson.Lau}@cityu.edu.hk

## Abstract

This paper explores the weakly-supervised referring image segmentation (WRIS) problem, and focuses on a challenging setup where target localization is learned directly from image-text pairs. We note that the input text description typically already contains detailed information on how to localize the target object, and we also observe that humans often follow a step-by-step comprehension process (*i.e.*, progressively utilizing target-related attributes and relations as cues) to identify the target object. Hence, we propose a novel Progressive Comprehension Network (PCNet) to leverage target-related textual cues from the input description for progressively localizing the target object. Specifically, we first use a Large Language Model (LLM) to decompose the input text description into short phrases. These short phrases are taken as target-related cues and fed into a Conditional Referring Module (CRM) in multiple stages, to allow updating the referring text embedding and enhance the response map for target localization in a multi-stage manner. Based on the CRM, we then propose a Region-aware Shrinking (RaS) loss to constrain the visual localization to be conducted progressively in a coarse-to-fine manner across different stages. Finally, we introduce an Instance-aware Disambiguation (IaD) loss to suppress instance localization ambiguity by differentiating overlapping response maps generated by different referring texts on the same image. Extensive experiments show that our method outperforms SOTA methods on three common benchmarks.

## 1   Introduction

Referring Image Segmentation (RIS) aims to segment a target object in an image via a user-specified input text description. RIS has various applications, such as text-based image editing [17, 13, 2] and human-computer interaction [62, 51]. Despite remarkable progress, most existing RIS works [7, 58, 27, 26, 21, 5] rely heavily on pixel-level ground-truth masks to learn visual-linguistic alignment. Recently, there has been a surge in interest in developing weakly-supervised RIS (WRIS) methods via weak supervisions, *e.g.*, bounding-boxes [9], and text descriptions [54, 18, 30, 4], to alleviate burden of data annotations. In this work, we focus on obtaining supervision from text descriptions only.

The relatively weak constraint of utilizing text alone as supervision makes visual-linguistic alignment particularly challenging. There are some attempts [30, 18, 22, 46] to explore various alignment workflows. For example, TRIS [30] classifies referring texts that describe the target object as positive texts while other texts as negative ones, to model a text-to-image response map for locating potential target objects. SAG [18] introduces a bottom-up and top-down attention framework to discover

---

[†] Joint corresponding authors.

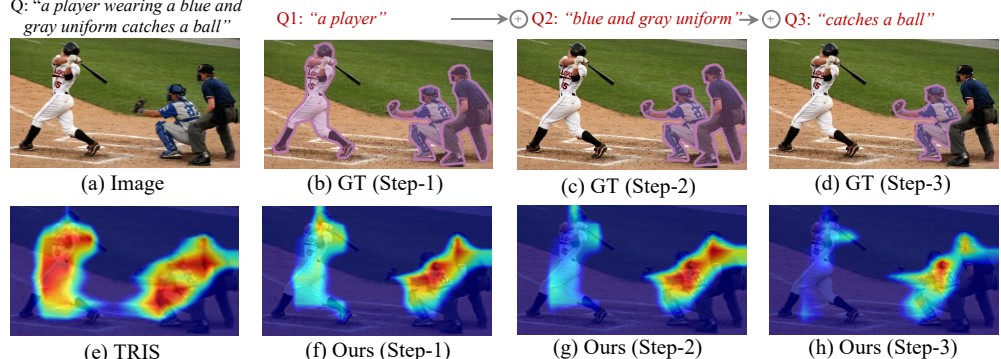

Figure 1: Given an image and a language description as inputs (a), RIS aims to predict the target object (d). Unlike existing methods (*e.g.*, TRIS [30] (e) – a WRIS method) that directly utilize the complete language description for target localization, we observe that humans would naturally break down the sentence into several key cues (*e.g.*, Q1 – Q3) and progressively converge onto the target object (from (b) to (d)). This behavior inspires us to develop the Progressive Comprehension Network (PCNet), which merges text cues pertinent to the target object step-by-step (from (f) to (h)), significantly enhancing visual localization. ⊕ denotes the text combination operation.

individual entities and then combine these entities as the target of the referring expression. However, these methods encode the entire referring text as a single language embedding. They can easily overlook some critical cues related to the target object in the text description, leading to localization ambiguity and even errors. For example, in Fig. 1(e), TRIS [30] erroneously activates all three players due to its use of cross-modality interactions between the image and the complete language embedding only.

We observe that humans typically localize target objects through a step-by-step comprehension process. Cognitive neuroscience studies [48, 41] also support this observation, indicating that humans tend to simplify a complex problem by breaking it down into manageable sub-problems and reasoning them progressively. For example, in Fig. 1(b-d), human perception would first begin with *"a player"* and identify all three players (b). The focus is then refined by the additional detail *"blue and gray uniform"*, which helps exclude the white player on the left (c). Finally, the action *"catches a ball"* helps further exclude the person on the right, leaving the correct target person in the middle (d).

Inspired by the human comprehension process, we propose in this paper a novel Progressive Comprehension Network (PCNet) for WRIS. We first employ a Large Language Model (LLM) [59] to dissect the input text description into multiple short phrases. These decomposed phrases are considered as target-related cues and fed into a novel Conditional Referring Module (CRM), which helps update the global referring embedding and enhance target localization in a multi-stage manner. We also propose a novel Region-aware Shrinking (RaS) loss to facilitate visual localization across different stages at the region level. ReS first separates the target-related response map (indicating the foreground region) from the target-irrelevant response map (indicating the background region), and then constrains the background response map to progressively attenuate, thus enhancing the localization accuracy of the foreground region. Finally, we notice that salient objects in an image can sometimes trigger incorrect response map activation for text descriptions that aim for other target objects. Hence, we introduce an Instance-aware Disambiguation (IaD) loss to reduce the overlapping of the response maps by rectifying the alignment score of different referring texts to the same object.

In summary, our main contributions are as follows :

- We propose the Progressive Comprehension Network (PCNet) for the WRIS task. Inspired by the human comprehension processes, this model achieves visual localization by progressively incorporating target-related textual cues for visual-linguistic alignment.

- Our method has three main technical novelties. First, we propose a Conditional Referring Module (CRM) to model the response maps through multiple stages for localization. Second, we propose a Region-aware Shrinking (RaS) loss to constrain the response maps across different stages for better cross-modal alignment. Third, to rectify overlapping localizations, we propose an Instance-aware Disambiguation (IaD) loss for different referring texts paired with the same image.

• We conduct extensive experiments on three popular benchmarks, demonstrating that our method outperforms existing methods by a remarkable margin.

## 2 Related work

**Referring Image Segmentation (RIS)** aims to segment the target object from the input image according to the input natural language expression. Hu *et al.* [14] proposes the first CNN-based RIS method. There are many follow-up works. Early methods [60, 28, 38] focus on object-level cross-modal alignment between the visual region and the corresponding referring expression. Later, many works explore the use of attention mechanisms [15, 7, 58, 19] or transformer architectures [58, 29] to model long-range dependencies, which can facilitate pixel-level cross-model alignment. For example, CMPC [15] employs a two-stage progressive comprehension model to first perceive all relevant instances through entity wording and then use relational wording to highlight the referent. In contrast, our approach leverages LLMs to decompose text descriptions into short phrases related to the target object, focusing on sentence-level (rather than word-level) comprehension, which aligns more closely with human cognition. Focusing on the visual grounding, DGA [56] also adopts multi-stage refinement. It aims to model visual reasoning on top of the relationships among the objects in the image. Differently, our work addresses the weakly RIS task and aims to alleviate the localization ambiguity by progressively integrating fine-grained attribute cues.

**Weakly-supervised RIS (WRIS)** has recently begun to attract some attention, as it can substantially reduce the burden of data labeling especially on the segmentation field [25, 57, 63]. Feng *et al.* [9] proposes the first WRIS method, which uses bounding boxes for annotations. Several subsequent works [18, 22, 30] attempt to use weaker supervision signal, *i.e.*, text descriptions. SAG [18] proposes to first divide image features into individual entities via bottom-up attention and then employ top-down attention to learn relations for combining entities. Lee *et al.* [22] generate Grad-CAM for each word of the description and then consider the relations using an intra-chunk and inter-chunk consistency. Instead of merging individual responses, TRIS [30] directly learns the text-to-image response map by contrasting target-related positive and target-unrelated negative texts. Inspired by the generalization capabilities of segmentation foundation models [20, 8, 34], PPT [6] enables effective integration with pre-trained language-image models [43, 33] and SAM [20] by a lightweight point generator to identify the referent and context noise. Despite their success, these methods encode the full text as a single embedding for cross-modality alignment, which overlooks target-related nuances in the textual descriptions. In contrast, our method proposes to combine progressive text comprehension and object-centric visual localization to obtain better fine-grained cross-modal alignment.

**Large Language Models (LLMs)** are revolutionizing various visual domains, benefited by their user-friendly interfaces and strong zero-shot prompting capabilities [3, 49, 1, 47]. Building on this trend, recent works [42, 55, 53, 45, 64] explore the integration of LLMs into vision tasks (*e.g.*, language-guided segmentation [55, 53], relation [23], and image classification [42]) through parameter-efficient fine-tuning or knowledge extraction. For example, LISA [55] and GSVA [53] utilize LLaVA [32], a large vision-language model (LVLM), as a feature encoder to extract visual-linguistic cross-modality features and introduce a small set of trainable parameters to prompt SAM [20] for reasoning segmentation. RECODE [23] and CuPL [42] leverage the knowledge in LLMs to generate informative descriptions as prompts for different categories classification. Unlike these works, we capitalize on the prompt capability of LLMs to help decompose a single referring description into multiple target object-related phrases, which are then used in our progressive comprehension process for RIS.

## 3 Our Method

In this work, we observe that when identifying an object based on a description, humans tend to first pinpoint multiple relevant objects and then narrow their focus to the target through step-by-step reasoning [48, 41]. Inspired by this, we propose a Progressive Comprehension Network (PCNet) for WRIS, which enhances cross-modality alignment by progressively integrating target-related text cues at multiple stages. Fig. 2 shows the overall framework of our PCNet.

Given an image $\mathbf{I}$ and a referring expression $\mathbf{T}$ as input, we first feed $\mathbf{T}$ into a Large Language Model (LLM) to break it down into $K$ short phrases $\mathcal{T}_{sub} = \{t_0, t_1, \cdots, t_{K-1}\}$, referred to as target-related text cues. We then feed image $\mathbf{I}$ and referring expression $\mathbf{T}$ and the set of short phrases $\mathcal{T}_{sub}$ into image encoder and text encoder to obtain visual feature $\mathbf{V}_0 \in \mathbb{R}^{H \times W \times C_v}$ and language

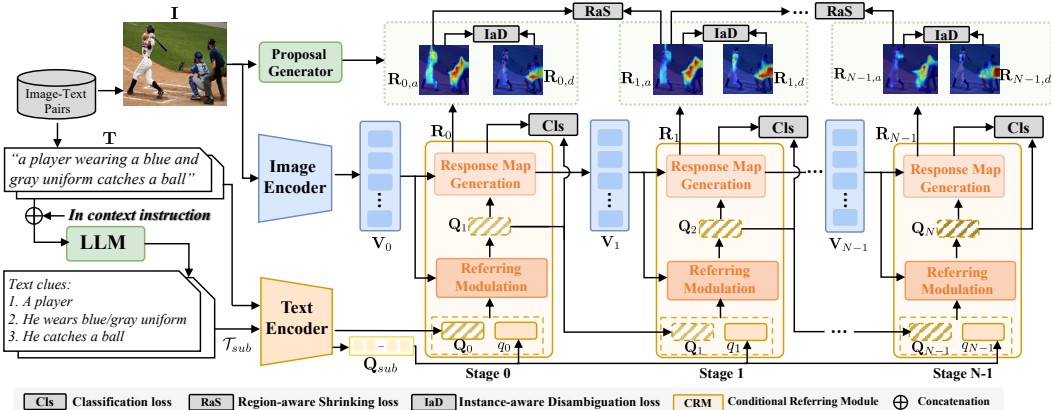

Figure 2: The pipeline of PCNet. Given a pair of image-text as input, PCNet enhances the visual-linguistic alignment by progressively comprehending the target-related textual nuances in the text description. It starts with using a LLM to decompose the input description into several target-related short phrases as target-related textual cues. The proposed Conditional Referring Module (CRM) then processes these cues to update the linguistic embeddings across multiple stages. Two novel loss functions, Region-aware Shrinking (RaS) and Instance-aware Disambiguation (IaD), are also proposed to supervise the progressive comprehension process.

feature $\mathbf{Q}_0 \in \mathbb{R}^{1 \times C_t}$, and $\mathcal{Q}_{sub} = \{\mathbf{q}_0, \mathbf{q}_1, \cdots, \mathbf{q}_{K-1}\}$, with $\mathbf{q}_k \in \mathbb{R}^{1 \times C_t}$, where $H = H_I/s$ and $W = W_I/s$. $C_v$ and $C_l$ denote the numbers of channels of visual and text features. $s$ is the ratio of down-sampling. We then use projector layers to transform the visual feature $\mathbf{V}_0$ and textual features $\mathbf{Q}_0$ and $\mathcal{Q}_{sub}$ to a unified dimension $C$, *i.e.*, $\mathbf{V}_0 \in \mathbb{R}^{H \times W \times C}$, $\mathbf{Q}_0$ and $\mathbf{q}_i$ are in $\mathbb{R}^{1 \times C}$.

We design PCNet with multiple consecutive Conditional Referring Modules (CRMs) to progressively locate the target object across $N$ stages[1]. Specifically, at stage $n$, the $n$-th CRM updates the referring embedding $\mathbf{Q}_n$ into $\mathbf{Q}_{n+1}$ conditioned on the short phrase $q_n$ from the proposed Referring Modulation block. Both $\mathbf{Q}_{n+1}$ and visual embedding $\mathbf{V}_n$ are fed into Response Map Generation to generate the text-to-image response map $\mathbf{R}_n$ and updated visual embedding $\mathbf{V}_{n+1}$. Finally, the response map $\mathbf{R}_{N-1}$ generated by the $n$-th CRM is used as the final localization result. To optimize the resulting response map for accurate visual localization, we employ the pre-trained proposal generator to obtain localization-matched mask proposals. We also propose Region-aware Shrinking (RaS) loss to constrain the visual localization in a coarse-to-fine manner, and Instance-aware Disambiguation (IaD) loss to suppress instance localization ambiguity.

In the following subsections, we first discuss how we decompose the input referring expression into target-related cues in Sec. 3.1. We then introduce the CRM in Sec. 3.2. Finally, we present our Region-aware Shrinking loss in Sec. 3.3, and Instance-aware Disambiguation loss in Sec. 3.4.

## 3.1 Generation of Target-related Cues

Existing works typically encode the entire input referring text description, and can easily overlook some critical cues (*e.g.*, attributes and relations) in the description (particularly for a long/complex description), leading to target localization problems. To address this problem, we propose dividing the input description into short phrases to process it individually. To do this, we leverage the strong in-context capability of the LLM [1] to decompose the text description. We design a prompt, with four parts, to instruct the LLM to do this: (1) general instruction $\mathbf{P}_G$; (2) output constraints $\mathbf{P}_C$; (3) in-context task examples $\mathbf{P}_E$; and (4) input question $\mathbf{P}_Q$. $\mathbf{P}_G$ describes the overall instruction, *e.g.*"decomposing the referring text into target object-related short phrases". $\mathbf{P}_C$ elaborates the output setting, *e.g.*, sentence length of each short phrase. In $\mathbf{P}_E$, we specifically curate several in-context pairs as guidance for the LLM to generate analogous outputs. Finally, $\mathbf{P}_Q$ encapsulates the input text description and the instruction words for the LLM to execute the operation. The process of generating

---
[1]Note that all counts start at 0.

target-related cues is formulated as:

$$\mathcal{T}_{sub} = \{t_0, t_1, \cdots, t_{K-1}\} = \text{LLM}\left(\mathbf{P}_G, \mathbf{P}_C, \mathbf{P}_E, \mathbf{P}_Q\right), \tag{1}$$

where $K$ represents the total number of phrases, which varies depending on the input description. Typically, longer descriptions more likely yield more phrases. To maintain consistency in our training dataset, we standardize it to five phrases (*i.e.*, $K = 5$). If fewer than five phrases are produced, we simply duplicate some of the short phrases to obtain five short phrases. In this way, phrases generated by LLM are related to the target object and align closely with our objective.

## 3.2   Conditional Referring Module (CRM)

Given the decomposed phrases (*i.e.*, target-related cues), we propose a CRM to enhance the discriminative ability on the target object region conditioned on these phrases, thereby improving localization accuracy. As shown in Fig. 2, the CRM operates across $N$ consecutive stages. At each stage, it first utilizes a different target-related cue to modulate the global referring embedding via a referring modulation block and then produces the image-to-text response map through a response map generation block.

**Referring Modulation Block.** Considering the situation at stage $n$, we first concatenate one target-related cue $q_n$ and the $L$ negative text cues obtained from other images [2], to form $\mathbf{q}'_n \in \mathbb{R}^{(L+1) \times C}$. We then fuse the visual features $\mathbf{V}_n$ with $\mathbf{q}'_n$ through a vision-to-text cross-attention, to obtain vision-attended cue features $\hat{\mathbf{q}}_n \in \mathbb{R}^{(L+1) \times C}$, as:

$$\boldsymbol{A}_{v \to t} = \text{SoftMax}\left((\mathbf{q}'_n W_1^{q'}) \otimes (\mathbf{V}_n W_2^V)^\top / \sqrt{C}\right); \ \hat{\mathbf{q}}_n = \text{MLP}(\boldsymbol{A}_{v \to t} \otimes (\mathbf{V}_n W_3^V)) + \mathbf{q}'_n, \tag{2}$$

where $\boldsymbol{A}_{v \to t} \in \mathbb{R}^{(L+1) \times H \times W}$ denotes the vision-to-text inter-modality attention weight. $W_*^V$ and $W_*^{q'}$ are learnable projection layers. $\otimes$ denotes matrix multiplication. Using the vision-attended cue features $\hat{\mathbf{q}}_n$, we then enrich the global textual features $\mathbf{Q}_n$ into cue-enhanced textual features $\mathbf{Q}_{n+1} \in \mathbb{R}^{1 \times C}$ through another text-to-text cross-attention, as:

$$\boldsymbol{A}_{t \to t} = \text{SoftMax}\left((\mathbf{Q}_n W_1^Q) \otimes (\hat{\mathbf{q}}_n W_2^{\hat{q}})^\top / \sqrt{C}\right); \ \mathbf{Q}_{n+1} = \text{MLP}(\boldsymbol{A}_{t \to t} \otimes (\hat{\mathbf{q}}_n W_3^{\hat{q}})) + \mathbf{Q}_n, \tag{3}$$

where $\boldsymbol{A}_{t \to t} \in \mathbb{R}^{1 \times (L+1)}$ represents the text-to-text intra-modality attention weight. $W_*^Q$ and $W_*^{\hat{q}}$ are learnable projection layers. In this way, we can enhance the attention of $\mathbf{Q}_n$ on the target object by conditioning its own target-related cue features and the global visual features.

**Response Map Generation.** To compute the response map, we first update the visual features $\mathbf{V}_n$ to $\hat{\mathbf{V}}_n$ by integrating them with the updated referring text embedding $\mathbf{Q}_{n+1}$ using a text-to-visual cross-attention, thereby reducing the cross-modality discrepancy. Note that $\hat{\mathbf{V}}_n$ is then used in the next stage (*i.e.*, $\mathbf{V}_{n+1} = \hat{\mathbf{V}}_n$). The response map $\mathbf{R}_n \in \mathbb{R}^{H \times W}$ at the $n$-th stage is computed as:

$$\mathbf{R}_n = \text{Norm}(\text{ReLU}(\hat{\mathbf{V}}_n \otimes \mathbf{Q}_{n+1}^\top)), \tag{4}$$

where Norm normalizes the output in the range of [0,1]. To achieve global visual-linguistic alignment, we adopt classification loss $\mathcal{L}_{\text{Cls}}$ in [30] to optimize the generation of the response map at each stage. It formulates the target localization problem as a classification process to differentiate between positive and negative text expressions. While the referring text expressions for an image are used as positive expressions, the ones from other images can be used as negative for this image. More explanations are given in appendix.

## 3.3   Region-aware Shrinking (RaS) Loss

Despite modulating the referring attention with the target-related cues stage-by-stage, image-text classification often activates irrelevant background objects due to its reliance on global and coarse response map constraints. Ideally, as the number of target-related cues used increases across each stage, the response map should become more compact and accurate. However, directly constraining the latter stage to have a more compact spatial activation than the former stage can lead to a trivial

---

[2]Refer to the Appendix for more details.

solution (*i.e.*, without target activation). To address this problem, we propose a novel region-aware shrinking (RaS) loss, which segments the response map into foreground (target) and background (non-target) regions. Through contrastive enhancement between these regions, our method gradually reduces the background interference while refining the foreground activation in the response map.

Specifically, at stage $n$, we first employ a pretrained proposal generator to obtain a set of mask proposals, $\mathcal{M} = \{m_1, m_2, \cdots, m_P\}$, where each proposal $m_p$ is in $\mathbb{R}^{H \times W}$ and $P$ is the total number of segment proposals. We then compute a alignment score between the response map $\mathbf{R}_n$ and each proposal $m_p$ in $\mathcal{M}$ as:

$$\mathcal{S}_n = \{s_{n,1}, s_{n,2}, \cdots, s_{n,P}\} \text{ with } s_{n,p} = \max(\mathbf{R}_n \odot m_p), \tag{5}$$

where $\odot$ denotes the hadamard product. The proposal with the highest score (denoted as $m_f$) is then treated as the target foreground region, while the combination of other proposals (denoted as $m_b$) is regarded as non-target background regions. With the separated regions, we define a localization ambiguity $S_n^{amb}$, which measures the uncertainty of the target object localization in the current stage $n$, as:

$$S_n^{amb} = 1 - \left(\text{IoU}(\mathbf{R}_n, m_f) - \text{IoU}(\mathbf{R}_n, m_b)\right), \tag{6}$$

where $S_n^{amb}$ is in the range of $[0, 1]$, and IoU denotes the intersection over union. When the localization result (*i.e.*, the response map) matches the only target object proposal instance exactly, ambiguity is 0. Conversely, if it matches the more background proposals, ambiguity approaches 1.

Assuming that each target in the image corresponds to an instance, by integrating more cues, the model will produce a more compact response map and gradually reduce the ambiguity. Consequently, based on the visual localization results from two consecutive stages, we can formulate the region-aware shrinking objective for a total of $N$ stages as:

$$\mathcal{L}_{\text{RaS}} = \frac{1}{N-1} \sum_{n=0}^{N-2} \max\left(0, (S_{n+1}^{amb} - S_n^{amb})\right). \tag{7}$$

By introducing region-wise ambiguity, $\mathcal{L}_{\text{RaS}}$ can direct non-target regions to converge towards attenuation while maintaining and improving the quality of the response map in the target region. This enables the efficient integration of target-related textual cues for progressively finer cross-modal alignment. Additionally, the mask proposals can also provide a shape prior to the target region, which helps to further enhance the accuracy of the target object localization.

### 3.4 Instance-aware Disambiguation (IaD) Loss

Although the RaS loss can help improve the localization accuracy by reducing region-wise ambiguity within one single response map, it takes less consideration of the relation between different instance-wise response maps. Particularly, we note that, given different referring descriptions that refer to different objects of an image, there are usually some overlaps among the corresponding response maps. For example, in Fig. 2, the player in the middle is simultaneously activated by two referring expressions (*i.e.*, the response maps $\mathbf{R}_{*,a}$ and $\mathbf{R}_{*,d}$ have overlapping activated regions), resulting in inaccurate localization. To address this problem, we propose an Instance-aware Disambiguation (IaD) loss to help enforce that different regions of the response maps within a stage are activated if the referring descriptions of an image refer to different objects.

Specifically, given a pair of image $\mathbf{I}_a$ and input text description $\mathbf{T}_a$, we first sample extra $N_d$ text descriptions, $\mathcal{T}_d = \{t_1, t_2, \cdots, t_{N_d}\}$, where the referred target object of each text description $t_d$ is in the image $\mathbf{I}_a$ but is different from the target object referred to by $\mathbf{T}_a$. We then obtain the image-to-text response maps $\mathbf{R}_a$ and $\mathcal{R}_d = \{\mathbf{R}_1, \mathbf{R}_2, \cdots, \mathbf{R}_{N_d}\}$ for $\mathbf{T}_a$ and $\mathcal{T}_d$ through Eq. (4). Here, we omit the stage index $n$ for clarity. Then, based on the Eq. (5), we obtain the alignment scores $\mathcal{S}_a$ and $\{\mathcal{S}_d\}_{d=1}^{N_d}$ for $\mathbf{T}_a$ and $\mathcal{T}_d$. In $\mathcal{S}$, the larger the value, the higher the alignment between the corresponding proposal (specified by the index) and the current text. To disambiguate overlapping activated regions, we constrain that the maximum index of the alignment score between $\mathcal{S}_a$ and each of $\mathcal{S}_d$ must be different from each other (*i.e.*, different texts must activate different objects). Here, we follow [50] to compute the index vector, $y \in \mathbb{R}^{1 \times P}$, as:

$$y = \texttt{one-hot}(\texttt{argmax}(\mathcal{S})) + \mathcal{S} - \texttt{sg}(\mathcal{S}), \tag{8}$$

Table 1: Quantitative comparison using mIoU and PointM metrics. "(U)" and "(G)" indicate the UMD and Google partitions. **"Segmentor"** denotes utilizing the pre-trained segmentation models (SAM [20] by default) for segmentation mask generation. † denotes that the method is fully-supervised. "–" means unavailable values. Oracle represents the evaluation of the best proposal mask based on ground-truth. Best and second-best performances are marked in **bold** and underlined.

| Metric | Method | Backbone | Segmentor | RefCOCO | | | RefCOCO+ | | | RefCOCOg | | |
|---|---|---|---|---|---|---|---|---|---|---|---|---|
| | | | | Val | TestA | TestB | Val | TestA | TestB | Val (G) | Val (U) | Test (U) |
| PointM↑ | GroupViT [54] | GroupViT | ✗ | 25.0 | 26.3 | 24.4 | 25.9 | 26.0 | 26.1 | 30.0 | 30.9 | 31.0 |
| | CLIP-ES [25] | ViT-Base | ✗ | 41.3 | 50.6 | 30.3 | 46.6 | 56.2 | 33.2 | 49.1 | 46.2 | 45.8 |
| | WWbL [46] | VGG16 | ✗ | 31.3 | 31.2 | 30.8 | 34.5 | 33.3 | 36.1 | 29.3 | 32.1 | 31.4 |
| | SAG [18] | ViT-Base | ✗ | 56.2 | 63.3 | 51.0 | 45.5 | 52.4 | 36.5 | 37.3 | – | – |
| | TRIS [30] | ResNet-50 | ✗ | 51.9 | 60.8 | 43.0 | 40.8 | 40.9 | 41.1 | 52.5 | 51.9 | 53.3 |
| | PCNet$_F$ | ResNet-50 | ✗ | 59.6 | 66.6 | 48.2 | 54.7 | 65.0 | 44.1 | 57.9 | 57.0 | 57.2 |
| | PCNet$_S$ | ResNet-50 | ✗ | **60.0** | **69.3** | **52.5** | **58.7** | **65.5** | **45.3** | **58.6** | **57.9** | **57.4** |
| mIoU↑ | LAVT† [58] | Swin-Base | N/A | 72.7 | 75.8 | 68.7 | 65.8 | 70.9 | 59.2 | 63.6 | 63.3 | 63.6 |
| | GroupViT [54] | GroupViT | ✗ | 18.0 | 18.1 | 19.3 | 18.1 | 17.6 | 19.5 | 19.9 | 19.8 | 20.1 |
| | CLIP-ES [25] | ViT-Base | ✗ | 13.8 | 15.2 | 12.9 | 14.6 | 16.0 | 13.5 | 14.2 | 13.9 | 14.1 |
| | TSEG [15] | ViT-Small | ✗ | 25.4 | – | – | 22.0 | – | – | 22.1 | – | – |
| | WWbL [46] | VGG16 | ✗ | 18.3 | 17.4 | 19.9 | 19.9 | 18.7 | 21.6 | 21.8 | 21.8 | 21.8 |
| | SAG [18] | ViT-Base | ✗ | **33.4** | 33.5 | **33.7** | 28.4 | 28.6 | **28.0** | 28.8 | – | – |
| | TRIS [30] | ResNet-50 | ✗ | 25.1 | 26.5 | 23.8 | 22.3 | 21.6 | 22.9 | 26.9 | 26.6 | 27.3 |
| | PCNet$_F$ | ResNet-50 | ✗ | 30.9 | 35.2 | 26.3 | 28.9 | 31.9 | 26.5 | 29.8 | 29.7 | 30.2 |
| | PCNet$_S$ | ResNet-50 | ✗ | 31.3 | **36.8** | 26.4 | **29.2** | **32.1** | 26.8 | **30.7** | **30.0** | **30.6** |
| | CLIP [43] | ResNet-50 | ✔ | 36.0 | 37.9 | 30.6 | 39.2 | 42.7 | 31.6 | 37.5 | 37.4 | 37.8 |
| | SAG [18] | ViT-Base | ✔ | 44.6 | 50.1 | 38.4 | 35.5 | 41.1 | 27.6 | 23.0 | – | – |
| | TRIS [30] | ResNet-50 | ✔ | 41.1 | 48.1 | 31.9 | 31.6 | 31.9 | 30.6 | 38.4 | 39.0 | 39.9 |
| | PPT [6] | ViT-Base | ✔ | 46.8 | 45.3 | 46.3 | 45.3 | 45.8 | 44.8 | 43.0 | – | – |
| | PCNet$_S$ | ResNet-50 | ✔ | **52.2** | **58.4** | 42.1 | **47.9** | **56.5** | **36.2** | **47.3** | **46.8** | **46.9** |
| | Oracle | ResNet-50 | ✔ | 72.7 | 75.3 | 67.7 | 73.1 | 75.5 | 68.2 | 69.0 | 68.3 | 68.4 |

where $\text{sg}(\cdot)$ represents the stop gradient operation. Finally, we denote the index vectors for $\mathcal{S}_a$ and $\{\mathcal{S}_d\}_{d=1}^{N_d}$ as $y_a$ and $\{y_d\}_{d=1}^{N_d}$, and we formulate the IaD loss as:

$$\mathcal{L}_{\text{IaD}} = \frac{1}{N_d} \sum_{d=1}^{N_d} \left(1 - ||y_a - y_d||^2\right). \tag{9}$$

By enforcing the constraint at each stage, the response maps activated by different referring descriptions in an image for different instances are separated, and the comprehension of the discriminative cues is further enhanced.

## 4 Experiments

### 4.1 Settings

**Dataset.** We have conducted experiments on three standard benchmarks: RefCOCO [61], RefCOCO+ [61], and RefCOCOg [39]. They are constructed based on MSCOCO [24]. Specially, the referring expressions in RefCOCO and RefCOCO+ focus more on object positions and appearances, respectively, and they are characterized by succinct descriptions, averaging 3.5 words in length. RefCOCOg contains much longer sentences (average length of 8.4 words), making it more challenging than others. RefCOCOg includes two partitions: UMD [40] and Google [39].

**Implementation Details.** We train our framwork for 15 epochs with a batch size of 36 on an RTX4090 GPU. The total loss for training is $\mathcal{L}_{\text{total}} = \mathcal{L}_{\text{Cls}} + \mathcal{L}_{\text{RaS}} + \mathcal{L}_{\text{IaD}}$. By default, we set the number of stages $N$ to 3, and the number of the additional text descriptions sampled for each image $N_d$ to 1. Without loss of generality, we use FreeSOLO [52] and SAM [20] as the proposal generators to obtain two versions: PCNet$_S$ and PCNet$_F$. Refer to Sec. A for more implementation details.

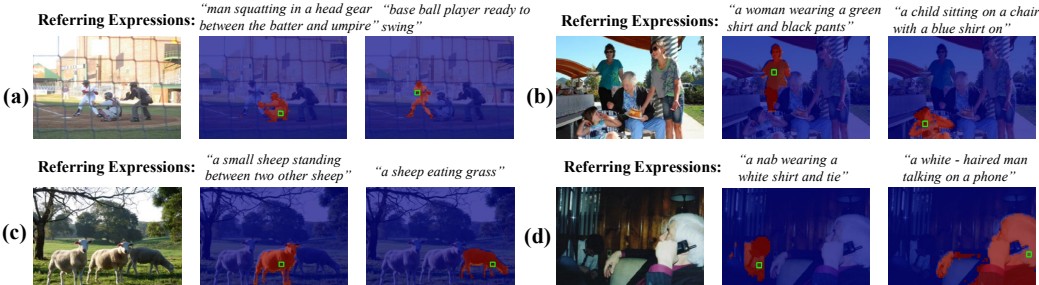

Figure 3: qVisual results of our PCNet. The green markers denote the peaks of the response maps.

**Evaluation Metrics.** We argue that the key to WRIS is target localization, and the performance evaluation should not rely primarily on pixel-wise metrics. With the accurate localization points, pixel-level masks can be readily obtained by prompting the pre-trained segmentors (*e.g.*, SAM [20]). Thus, following [18, 22, 30], we adopt localization-based metric (*i.e.*, PointM), and pixel-wise metrics (*i.e.*, mean and overall intersection-over-union (mIoU and oIoU) for evaluation. PointM [30] is used to evaluate the localization accuracy, which computes the ratio of activation peaks in the mask region.

## 4.2 Comparison with State-of-the-Art Methods

**Quantitative comparison.** Tab. 1 compares our method with various SOTA methods. Specifically, we first compare target localization accuracy using the PointM metric. We evaluate two model variants: $PCNet_F$ and $PCNet_S$, which use different segmentors (*e.g.*, FreeSOLO [52] and SAM [20]) to extract mask proposals for RaS and IaD losses. Even when using FreeSOLO as the proposal generator, our model still significantly outperforms all compared methods. For example, on the most challenging dataset, RefCOCOg, with more complex object relationships and longer texts, $PCNet_F$ achieves performance improvements of 55.2% and 10.3% on the Val (G) set compared to SAG and TRIS[3]. $PCNet_S$ further boosts the performance if we replace FreeSOLO with the stronger SAM.

In addition, we verify the accuracy of the response map through pixel-wise mIoU metric. Results are shown in the middle part of Tab. 1. Our PCNet still achieves superior performances on all benchmarks, against all compared methods. Particularly, $PCNet_F$ and $PCNet_S$ outperform TRIS by an improvement of 10.8% and 14.1% mIoU, respectively, on the RefCOCOg Val (G) set. In the bottom part of Tab. 1, we compare the accuracy of the extracted mask proposals generated using the target localization point (*i.e.*, the peak point of the response map) to prompt SAM. We can see that our PCNet significantly outperforms other WRIS methods. We can also see that higher PointM values correlate with higher mIoU accuracy values of the corresponding mask proposals for different methods. We further tested the mask accuracy using the Ground-Truth localization point (*i.e.*, the last row), and find that its performance even surpasses the fully-supervised method, LAVT [58]. All these results highlight the critical importance of target localization (*i.e.*, peak point) for the WRIS task.

In Fig. 3, we show some visual results of our method across different scenes by using the target localization point (*i.e.*, the green marker) to prompt SAM to generate the target mask. Our method effectively localizes the target instance among other instances within the image, even in complex scenarios with region occlusion (a), multiple instances (b), similar appearance (c), and dim light (d).

## 4.3 Ablation Study

We conduct ablation experiments on the RefCOCOg dataset and report the results on the Val (G) set from both $PCNet_S$ and $PCNet_F$ in Tab. 2, and from $PCNet_F$ in Tab. 3, Tab. 4, and Tab. 5.

**Component Analysis.** In Tab. 2, we first construct a single-stage baseline (*1st* row) to optimize visual-linguistic alignment by removing all proposed components and using only the global image-text classification loss $\mathcal{L}_{\texttt{Cls}}$. We then introduce the proposed conditional referring module (CRM) to the baseline to allow for multi-stage progressive comprehension (*2nd* row). To validate the efficacy of the region-aware shrinking loss (RaS) and instance-aware disambiguation loss (IaD), we introduce them separately (*3rd* and *4th* rows). Finally, we combine all proposed components (*5th* row).

---

[3]For a fair comparison, we remove its 2nd stage as it is used for enhancing pixel-wise mask accuracy.

Table 2: Component ablations on RefCOCOg Val (G) set.

| $\mathcal{L}_{\texttt{Cls}}$ | CRM | $\mathcal{L}_{\texttt{RaS}}$ | $\mathcal{L}_{\texttt{IaD}}$ | PointM | | mIoU | | oIoU | |
|---|---|---|---|---|---|---|---|---|---|
| | | | | $PCNet_S$ | $PCNet_F$ | $PCNet_S$ | $PCNet_F$ | $PCNet_S$ | $PCNet_F$ |
| ✓ | | | | 51.7 | | 25.3 | | 25.1 | |
| ✓ | ✓ | | | 53.3 | | 26.8 | | 26.7 | |
| ✓ | ✓ | ✓ | | 57.7 | 56.4 | 29.8 | 28.5 | 29.6 | 28.5 |
| ✓ | ✓ | | ✓ | 55.3 | 54.3 | 28.3 | 27.7 | 28.2 | 27.8 |
| ✓ | ✓ | ✓ | ✓ | 58.6 | 57.9 | 30.7 | 29.8 | 30.6 | 30.1 |

Table 3: Ablation of the number of iterative stages $N$.

| $N$ | mIoU | oIoU | PointM |
|---|---|---|---|
| 1 | 27.4 | 27.3 | 55.3 |
| 2 | 29.3 | 29.4 | 57.3 |
| **3** | 29.8 | 30.1 | 57.9 |
| 4 | 29.5 | 29.8 | 56.7 |

Table 4: Ablation of different modulation strategies in CRM.

| Method | mIoU | oIoU | PointM |
|---|---|---|---|
| ADD | 28.5 | 28.4 | 56.3 |
| TTA | 29.3 | 29.1 | 57.1 |
| VTA+ADD | 29.2 | 29.1 | 57.2 |
| **VTA+TTA** | 29.8 | 30.1 | 57.9 |

Table 5: Ablation of the numbers of descriptions $N_d$ in IaD.

| $N_d$ | mIoU | oIoU | PointM |
|---|---|---|---|
| 0 | 28.5 | 28.5 | 56.4 |
| **1** | 29.8 | 30.1 | 57.9 |
| 2 | 29.8 | 29.7 | 57.8 |
| 3 | 29.7 | 29.6 | 57.7 |

The results demonstrate that ❶ even using only $\mathcal{L}_{\texttt{Cls}}$, progressively introducing target-related cues through CRM can still significantly enhance target object localization. In particular, $PCNet_S$ achieves improvements of 3.1% on PointM and 5.9% on mIoU; ❷ by using $\mathcal{L}_{\texttt{RaS}}$ to constrain response maps, making them increasingly compact and complete during the progressive comprehension process, the accuracy of target localization is dramatically enhanced, resulting in an improvement of 11.6% on PointM. ❸ although $\mathcal{L}_{\texttt{IaD}}$ can facilitate the separation of overlapping response maps between different instances within the same image and improve the discriminative ability of our model on the target object, the lack of constraints between consecutive stages results in a smaller performance improvement than $\mathcal{L}_{\texttt{RaS}}$; and ❹ all components are essential for our final PCNet, and combining them achieves the best performance. In Fig. 4, we also provide the visual results of the ablation study on two examples. We can see that each component can bring obvious localization improvement.

**Number of Iterative Stages.** In Tab. 3, we analyze the effect of the number of iterative stages $N$. When $N = 1$, we can only apply $\mathcal{L}_{\texttt{Cls}}$ and $\mathcal{L}_{\texttt{IaD}}$, but not $\mathcal{L}_{\texttt{RaS}}$, resulting in inferior results. Increasing $N$ from 1 to 2 significantly improves the performance due to the progressive introduction of target-related cues. However, the improvement from $N = 2$ to $N = 3$ is less pronounced than from $N = 1$ to $N = 2$, and the performance stabilizes at $N = 3$. At $N = 4$, the performance slightly declines. This is because when the effective short-phrases decomposed by LLM are fewer than the number of stages, we need to repeat text phrases in later stages, which may affect the loss optimization.

**Modulation Strategy.** In Tab. 4, we ablate different variants of CRM: ❶ directly adding target cue features $\mathbf{q}_n$ and global referring features $\mathbf{Q}_n$ (denoted as ADD); ❷ fusing $\mathbf{q}_n$ and $\mathbf{Q}_n$ using only text-to-text cross-attention (denoted as TTA); ❸ first employing a vision-to-text cross-attention to fuse visual features $\mathbf{V}_n$ and $\mathbf{q}_n$ to obtain vision-attended features $\hat{\mathbf{q}}_n$, and then adding them to $\mathbf{Q}_n$ (denoted VTA+ADD). The results demonstrate that ADD is the least efficient method. TTA

Q: "*a light brown color sweet vada with dark brown one next to it*"

Q: "*a catcher rushing to make a play on the ball*"

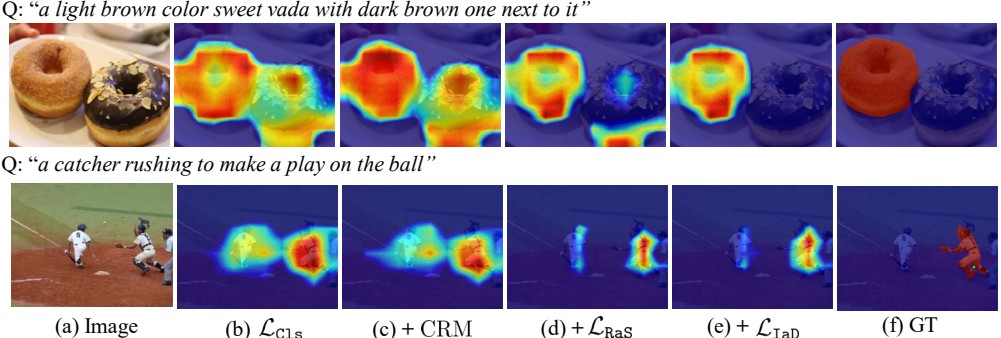

(a) Image    (b) $\mathcal{L}_{\texttt{Cls}}$    (c) + CRM    (d) + $\mathcal{L}_{\texttt{RaS}}$    (e) + $\mathcal{L}_{\texttt{IaD}}$    (f) GT

Figure 4: Visualization of the ablation study to show the efficacy of each proposed component.

*Q: "3 teddy bears sitting on a bed"*

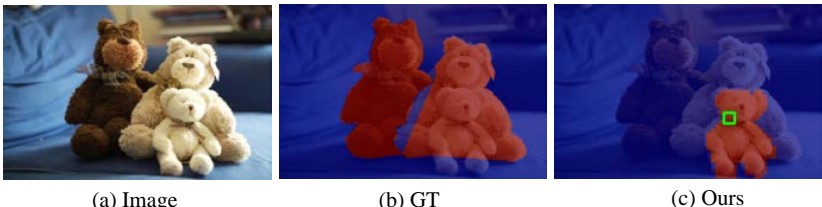

| (a) Image | (b) GT | (c) Ours |

Figure 5: A failure case of our PCNet. As our model design assumes that there is only one object referred to by the language expression, it usually returns only one object.

outperforms ADD but is less effective than VTA+ADD, verifying the importance of the vision context. Finally, our CRM combines VTA and TTA and achieves the best results.

**Number of Referring Texts.** In Tab. 5, we analyze the effect of $N_d$ used in $\mathcal{L}_{\text{LaD}}$. The results show that $N_d = 1$ is enough, and the performance deteriorates as $N_d$ increases. This is because an image typically has 2-3 text descriptions, which means $N_d$ should be 1-2. As $N_d$ increases, repeated sampling becomes more frequent, affecting model training and thus leading to poorer results.

## 5    Conclusion

In this paper, we have proposed a novel Progressive Comprehension Network (PCNet) to perfom progressive visual-linguistic alignment for the weakly-supervised referring image segmentation (WRIS) task. PCNet first leverages a LLM to decompose the input referring description into several target-related phrases, which are then used by the proposed Conditional Referring Module (CRM) to update the referring text embedding stage-by-stage, thus enhancing target localization. In addition, we proposed two loss functions, region-aware shrinking loss and instance-aware disambiguation loss, to facilitate comprehension of the target-related cues progressively. We have also conducted extensive experiments on three RIS benchmarks. Results show that the proposed PCNet achieves superior visual localization performances and outperforms existing SOTA WRIS methods by large margins.

Our method does have limitations. For example, as shown in Fig. 5, when the text description refers to multiple objects, our method fails to return all referring regions. This is because our model design always assumes that there is only one object referred to by the language expression. In the future, we plan to incorporate more fine-grained vision priors [35, 44] and open-world referring descriptions (*e.g.*, camouflaged [12], semi-transparent [31], shadow [36] and *etc.*) into the model design to enable a more generalized solution.

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

# A  More Implementation Details

## A.1  Generation of Target-related Cues

To obtain multiple target-related cues, we leverage the strong in-context capability of the Large Language Model (LLM) [16] to decompose the input referring expression and obtain the target-related textual cues. The Fig. 6 presents the LLM prompting details.

The prompt includes four parts: (1) general instruction $\mathbf{P}_G$, (2) output constraints $\mathbf{P}_C$, (3) in-context task examples $\mathbf{P}_E$, and (4) input question $\mathbf{P}_Q$. In part $\mathbf{P}_G$, we define a overall instruction for our task (i.e, decomposing the referring text) Then in part $\mathbf{P}_C$, we elaborate some details about the output (e.g, the sentence length for each cue description). In part $\mathbf{P}_E$, we curate

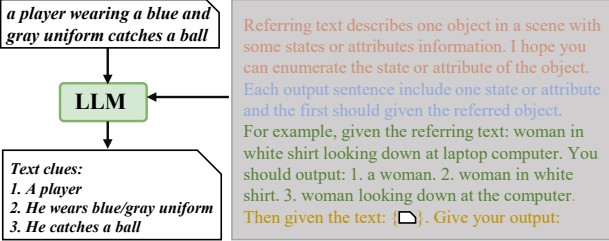

Figure 6: Flow of LLM-based referring text decomposition.

several in-context learning examples as guidance for the LLM to generating analogous output. Considering that the input referring expressions contain various sentence structures, in part $\mathbf{P}_E$ the more examples given, the more reliable the output will be. The part $\mathbf{P}_Q$ instructs the LLM to output the results given the input referring expression.

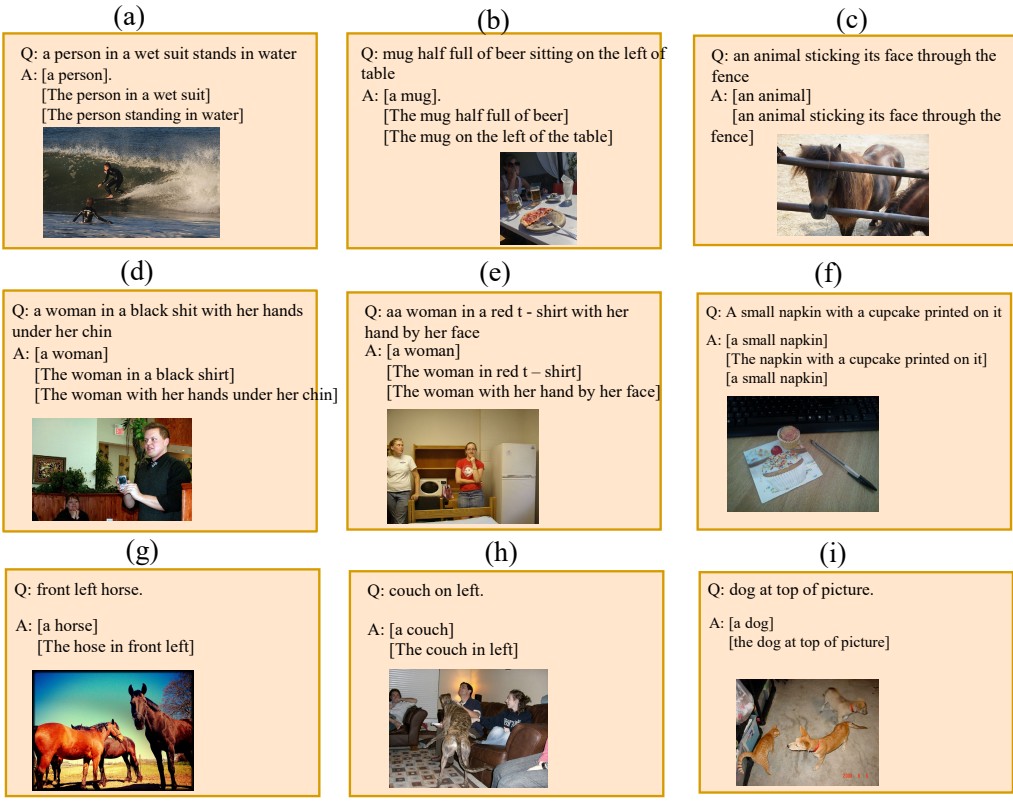

Figure 7: LLM generated examples. We show the LLM generated examples for long language expressions in (a)-(f) and the ones for short language expressions in (g)-(i). "Q" denotes the input language expression and the "A" denotes the output target-related textual cues of LLM.

In Fig. 7, we give some examples of decomposing the referring expressions. The "Q" denotes the input referring content and the "A" denotes the answers of LLM. In most cases, we can obtain reliable target-related textual cues that do not contradict the original text input. Besides, we also notice that there are some cases in which the text is not sufficiently decomposed (e.g, the example (c)) or the LLM outputs redundant results (e.g, the example (f)), which hinders the model to benefit from progressive comprehension to some extent.

## A.2 Text-to Image Classification Loss

Our work consists of multiple stages and utilizes $\mathcal{L}_{\texttt{cls}}$ in TRIS [30] at each stage independently for response maps optimization. Here, we omit the index of stage $n$ for clarity. $\mathcal{L}_{\texttt{cls}}$ formulates the target localization problem as a classification process to differentiate between positive and negative text expressions. The key idea of $\mathcal{L}_{\texttt{cls}}$ loss function is to contrast image-text pairs such that correlated image-text pairs have high similarity scores and uncorrelated image-text pairs have low similarity scores. While the referring text expressions for an image are used as positive expressions, the referring text expressions from other images can be used as negative expressions for this image. Thus, given a batch (i.e., $B$) of image samples , each sample is mutually associated with one positive reference text (i.e., a text describing a specific object in the current image) and mutually exclusive with L negative reference texts (texts that are not related to the target object in the image). Note that the number of batches is equal to the sum of the positive samples and the negative samples (i.e., $B = 1 + L$).

Speficially, in each training batch, $B$ image-text pairs $\{\mathbf{I}_i, \mathbf{T}_i\}_{i=1}^{B}$ are sampled. Through the language and vision encoders, we can get referring embeddings $\mathbf{Q} \in \mathbb{R}^{B \times C}$ and image embeddings $\mathbf{V} \in \mathbb{R}^{B \times H \times W \times C}$. Then, we obtain the response maps $\mathbf{R} \in \mathbb{R}^{B \times B \times H \times W}$ by applying cosine similarity calculation and normalization operation. After the pooling operation as done in TRIS, we further obtain the alignment score matrix $\mathbf{y} \in \mathbb{R}^{B \times B}$. According to the $\mathcal{L}_{\texttt{cls}}$, for $i_{\text{th}}$ image in the batch, there is a prediction score $\mathbf{y}[i, :]$, where $\mathbf{y}[i, i]$ predicted by the corresponding text deserves a higher value (i.e, the positive one) and the others deserve lower values ($L$ negative ones). Then classification loss for the $i_{th}$ image from the batch can be formulated as cross-entropy loss:

$$\mathcal{L}_{\texttt{cls},i} = -\frac{1}{B}\sum_{j=1}^{B}\left(\mathbb{1}_{i=j}\log\left(\frac{1}{1+e^{-\mathbf{y}[i,j]}}\right) + (1 - \mathbb{1}_{i=j})\log\left(\frac{e^{-\mathbf{y}[i,j]}}{1+e^{-\mathbf{y}[i,j]}}\right)\right),$$

and the classification loss for the batch can be formulated as:

$$\mathcal{L}_{\texttt{cls}} = \frac{1}{B}\sum_{i=1}^{B}\mathcal{L}_{\texttt{cls},i}$$

The $i$ denotes the index for the visual image and the $j$ denotes the index for the referring text.

## A.3 Referring Modulation Block

In Sec. 3.2, we have mentioned that the conditional referring module (CRM) utilizes the decomposed textual cues to progressively modulate the referring embedding via a modulation block across $N$ consecutive stages, and then produces the image-to-text response map by computing the patch-based similarity between visual and language embeddings. Specifically, the modulation block is implemented by a vision-to-text and a text-to-text cross-attention mechanism in cascade for facilitating the interaction between cross-modal features. In Fig. 8, we give an overview of the block design. For the block at each stage, We concatenate one target-related cue and the $L$ negative text cues obtained from other images as the conditional text cues and then obtain the vision-attended cue features by the vision-to-text attention. Then by learning the interaction between referring embedding and different textual cue embeddings, the block is expected to enhance the integration of discriminative cues.

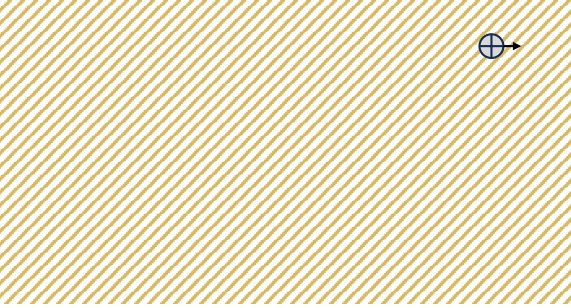

Figure 8: Illustration of referring modulation block.

## A.4 Training and Inference

We implement our framework on PyTorch and train it for 15 epochs with a batch size of 36 (*i.e.*, $L + 1$) on a RTX4090 GPU with 24GB of memory. The input images are resized to $320 \times 320$. We use ResNet-50 [11] as our backbone of image encoder, and utilize the pre-trained CLIP [43] model to initialize the image and text encoders. The down-sampling ratio of visual feature $s = 32$, the channels of vision feature $C_v = 2048$, text features $C_l = 1024$, and the unified hidden dimension $C = 1024$. The network is optimized using the AdamW optimizer [37] with a weight decay of $1e^{-2}$ and an initial learning rate of $5e^{-5}$ with polynomial learning rate decay. For the LLM, we utilize the open-source powerful language model Mistral 7B [16] for referring text decomposition. For the proposal generator, we set the number of extracted proposals $P = 40$ for each image.

# B  More Quantitative Studies

## B.1  Comparisons with other SOTA methods

Table 6: Different criterions for alignment score measurement in $\mathcal{L}_{\mathrm{RaS}}$.

| Alignment Score | mIoU | PointM | oIoU |
|---|---|---|---|
| Max | 29.8 | 57.9 | 30.1 |
| Avg | 29.1 | 56.4 | 29.2 |

Table 7: Different criterions for alignment score measurement in $\mathcal{L}_{\mathrm{IaD}}$.

| Alignment Score | mIoU | PointM | oIoU |
|---|---|---|---|
| Max | 29.8 | 57.9 | 30.1 |
| Avg | 28.8 | 54.9 | 29.0 |

In Tab. 6 and Tab. 7, we conduct the ablation studies about the measurement criterion of alignment score. The results demonstrate that the maximum value of the response map in each proposal better represents the alignment level of region-wise cross-modal alignment than the average value. To validate the effectiveness of the modeling the progressive comprehension, we also quantitatively compare the outputs of our method at different stages in Tab. 8. The results show that the localization results gradually improve with more discriminative cues integration, especially in the early stages.

Table 8: Comparison between different stages.

| Stage Num. | Stage 0 | Stage 1 | Stage 2 |
|---|---|---|---|
| mIoU | 28.6 | 29.4 | 29.8 |
| PointM | 56.7 | 57.6 | 57.9 |

## B.2  More Ablation Studies

**Comparison between IaD loss and others.** In our IaD loss $\mathcal{L}_{\mathrm{IaD}}$, we adopt a hard assignment for deriving the loss function as GroupViT [54]. The motivation is that we aims to get the pseudo mask prediction by the accurate peak value point (i.e., the hard assignment results) instead of relying on whole score distribution $S(\cdot)$ (e.g., $S_a$, $S_d$ in Sec. 3.4). Thus utilizing the hard assignment to derive the IaD loss well matches our purpose, which helps rectify the ambiguous localization results. If we use the soft assignment (e.g., measuring KL divergence between $S_a$ and $S_d$), though the equivalent may be simpler, it not only does not match our purpose but also introduces more tricky components for optimization (e.g., extra distribution regularization is required). In order to verify the argument, we conduct a comparison on RefCOCOg(G) val dataset as Tab. 9. The $\mathcal{L}_{\mathrm{IaD}}$ even causes a slight decline, while the proposed loss $\mathcal{L}_{\mathrm{KL}}$ brings clear improvement on localization accuracy.

Table 9: Comparison between IaD loss and KL loss.

| $\mathcal{L}_{\mathrm{CLs}}$ | $\mathcal{L}_{\mathrm{KL}}$ | $\mathcal{L}_{\mathrm{IaD}}$ | PointM | mIoU |
|---|---|---|---|---|
| ✓ | | | 51.7 | 25.3 |
| ✓ | ✓ | | 51.2 | 24.8 |
| ✓ | | ✓ | 53.1 | 26.6 |

**Comparison between IaD loss and calibration loss in TRIS.** There are essential differences between them. First, the calibration loss in TRIS [30] is used to suppress noisy background activation and thus help to re-calibrate the target response map. In contrast, in our method, we observe that, multiple referring texts corresponding to different instances in one image may locate the same instances (or we say overlapping), due to the lack of instance-level supervision in WRIS.

As for the implementation, the calibration loss adopts the global CLIP score of image-text to implement a simple contrastive learning for revising the response map. Differently, we simultaneously infer the response maps of different referring texts from the same image, and obtain the instance-level localization results by choosing the mask proposal with max alignment score. To further verify the superiority of our loss, we conduct an ablation on the RefCOCO(val) dataset. We use the TRIS without calibration loss as the baseline and then separately introduced these two loss functions for comparison. Both ablations demonstrate that the IaD loss not only refines the response map (mIoU) but also significantly improves the localization accuracy (PointM).

Table 10: Comparison between IaD loss and calibration loss $\mathcal{L}_{\text{Cal}}$ .

| $\mathcal{L}_{\text{CLs}}$ | $\mathcal{L}_{\text{IaD}}$ | $\mathcal{L}_{\text{Cal}}$ | PointM | mIoU |
|:---:|:---:|:---:|:---:|:---:|
| ✓ | | | 50.3 | 24.6 |
| ✓ | ✓ | | 51.4 | 26.4 |
| ✓ | | ✓ | 54.7 | 26.3 |

## C  More Visualization of Localization Results

**Progressive Comprehension for Localization.** In Fig. 9, we also give visualization of each stage's response map for qualitative analysis. The results show that our proposed CRM module can effectively integrate the target-related textual cues. For example, in the first row, the method produces ambiguous localization result at the first stage. After taking the cue *"with gold necklace"* into consideration, the attention is transferred to the target object at the second stage. Finally, after considering all the cues, the method produces less ambiguous and more accurate localization results.

Q: *woman with gold necklace sitting behind little birthday girl*

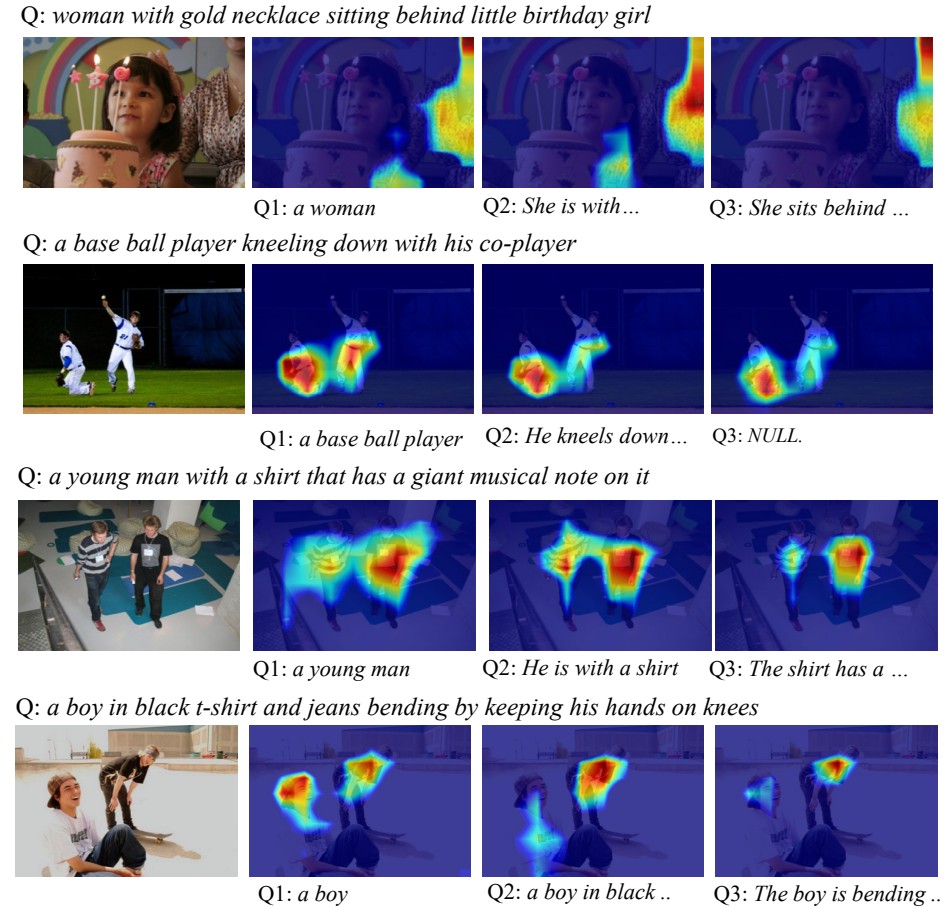

Q1: *a woman*    Q2: *She is with…*    Q3: *She sits behind …*

Q: *a base ball player kneeling down with his co-player*

Q1: *a base ball player*    Q2: *He kneels down…*    Q3: *NULL.*

Q: *a young man with a shirt that has a giant musical note on it*

Q1: *a young man*    Q2: *He is with a shirt*    Q3: *The shirt has a …*

Q: *a boy in black t-shirt and jeans bending by keeping his hands on knees*

Q1: *a boy*    Q2: *a boy in black ..*    Q3: *The boy is bending ..*

Figure 9: Visualization of progressive localization. With the integration of discriminative cues, the identification of target instance gradually improves.

**Qualitative Comparison with Other method.** More qualitative comparisons of our method with other methods are shown in the Fig. 10. For the example shown in the fifth row, the query is "a man in a arm striped sweater". The TRIS [30] mistakenly locates the left man as the target regions. In contrast, our PCNet optimizes the response map generation process by continuously modulating the referring embedding query conditioned on the target-related cues instead of a static referring embedding. As a result, our method can obtain more accurate localization result.

## D    Proposal Generator

In this work, we adopt the two representative pre-trained segmentors: FreeSOLO [52] and SAM [20] for proposal generation. Specifically, the FreeSOLO [52] is a fully unsupervised learning method that learns class-agnostic instance segmentation without any annotations. The SAM's training utilizes densely labeled data, but it does not include semantic supervision. This supervision does not contradict our weakly supervised RIS setting. More importantly, it offers a promising solution as an image segmentation foundation model and can be used for refining the coarse localization results from weakly-supervised methods into precise segmentation masks as done in the recent works [10, 63]. For the usage of SAM, We adopt the ViT-H backbone, the hyperparameter *predicted iou threshold* and *stability score threshold* are set to 0.7, and *points per side* is set to 8.

In Fig. 11 and Fig. 12, we give the generated mask proposals examples by FreeSOLO [52] and SAM [20], respectively. We notice that there are often overlaps among the generated proposals. Thus, we refine the generated proposals by filtering out candidate proposals with small area (the threshold is set as 1000) and then selecting the ones with smaller intersection over union (the threshold is set as 0.8). Considering that the number of proposals generated by the segmentor may be different for different image inputs, in implementation, we maintain consistency by selecting the top 40 proposals with the largest area ($P = 40$). If fewer than 40, we simply complete it with an all-zero mask.

## E    Broader Impacts

While we do not foresee our method causing any direct negative societal impact, it may potentially be leveraged by malicious parties to create applications that could misuse the segmentation capabilities for unethical or illegal purposes. We urge the readers to limit the usage of this work to legal use cases.

Q: "*the guy wearing white sitting on the couch watching his friends play video games*"

Q: "*a catcher rushing to make a play on the ball*"

Q: "*a boy in black t - shirt and jeans bending by keeping his hands on knees*"

Q: "*a smiling man with a small infant wearing a beige t - shirt and gray jeans*"

Q: "*a man in a arm striped sweater*"

Q: "*a postcard with picture of face of cute girl*"

Q: "*dark haired woman wearing a blue jacket next to a teddy bear*"

| **(a) Image** | **(b) TRIS** | **(c) SAG** | **(d) PCNet** | **(e) GT** |

Figure 10: More visual comparison between our method with TRIS and SAG for WRIS.

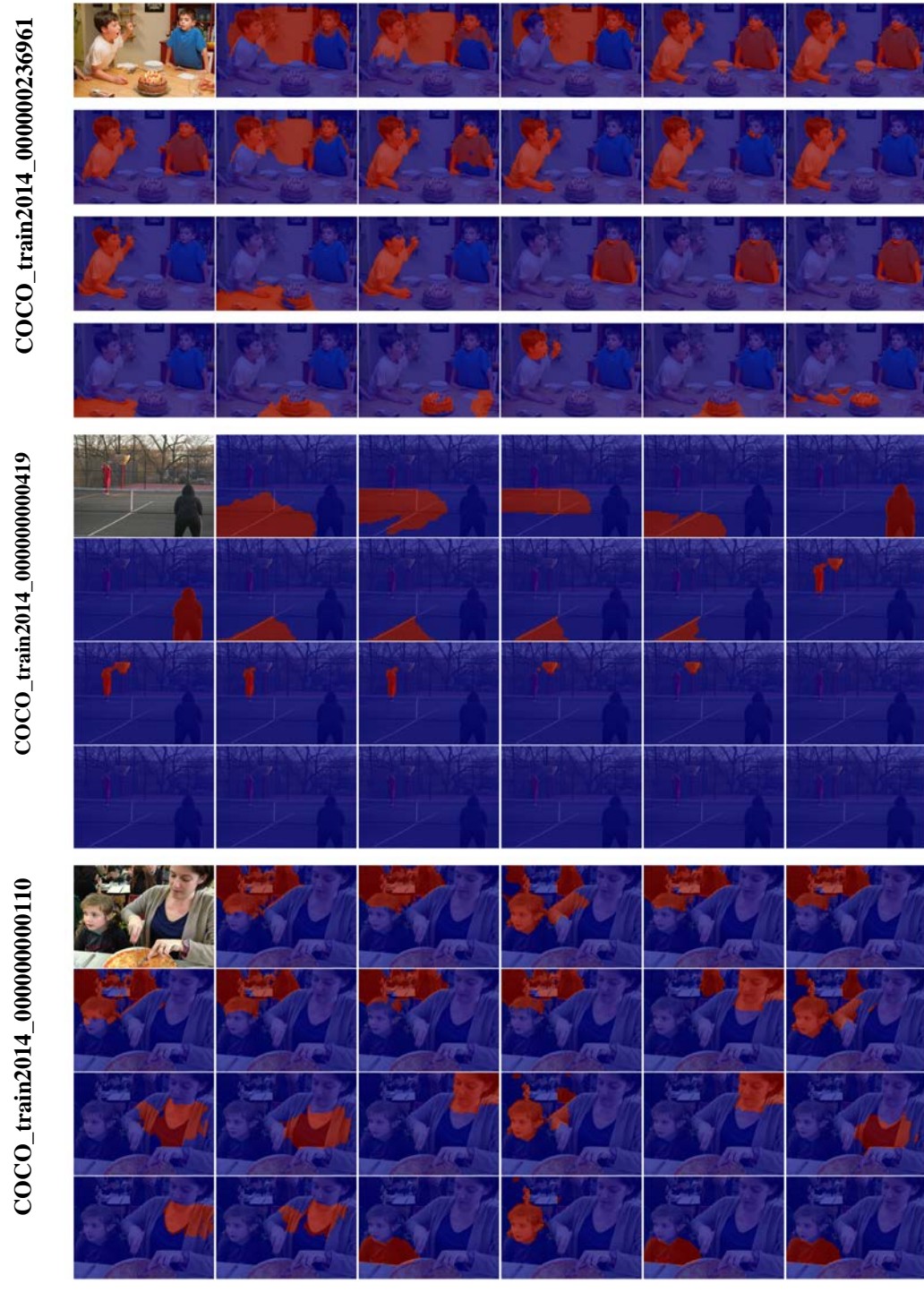

Figure 11: FreeSOLO [52] generated mask proposals examples.

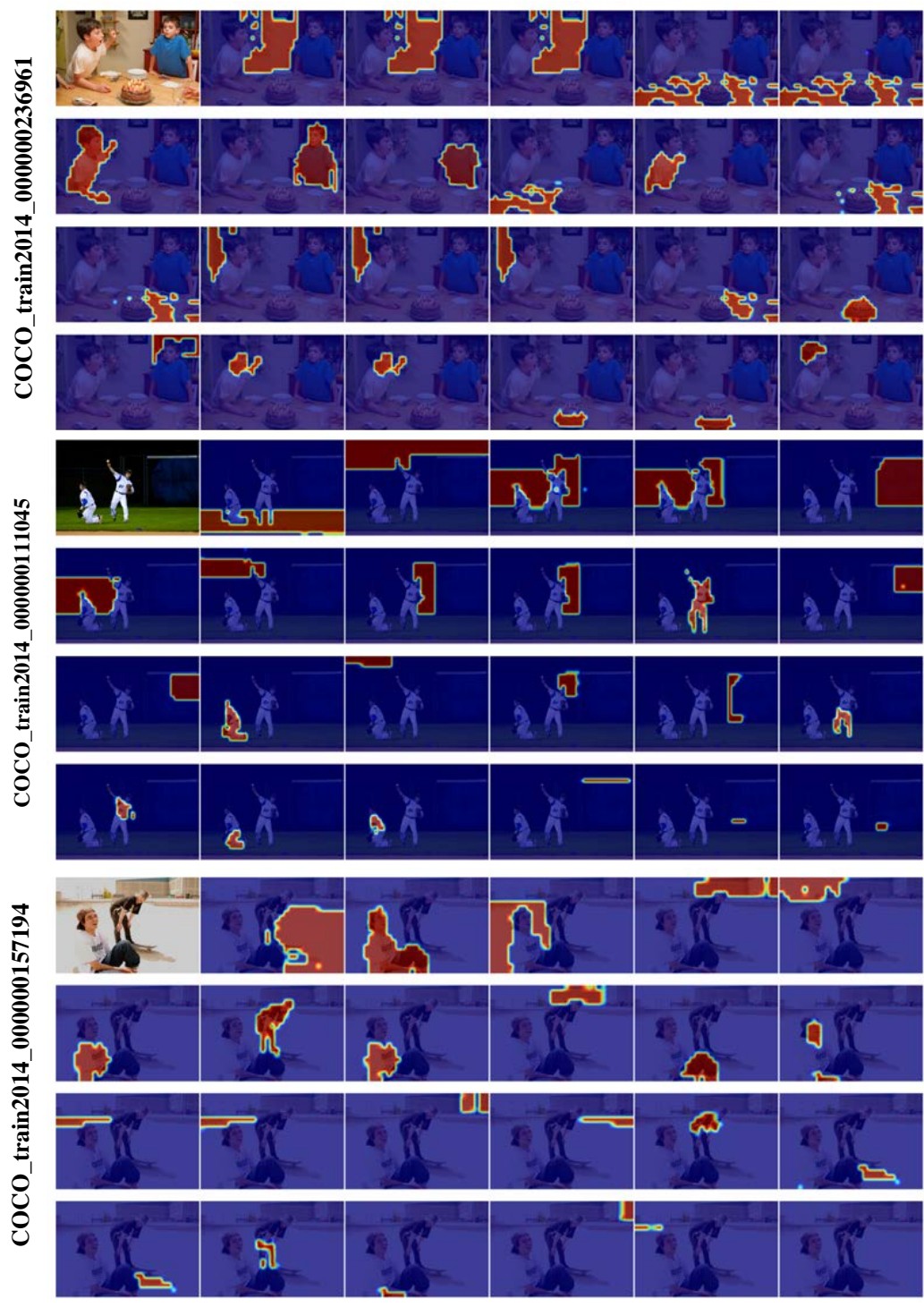

Figure 12: SAM [20] generated mask proposals examples.

