# OpenReview forum: "Boosting Weakly Supervised Referring Image Segmentation via Progressive Comprehension"
_NeurIPS.cc/2024/Conference — NeurIPS 2024 poster_

### Official Review · Reviewer_K7SG · 2024-07-02

**Soundness:** 4
**Presentation:** 4
**Contribution:** 4
**Rating:** 7
**Confidence:** 4

**Summary:**

Aiming for the WRIS task, the work proposes a novel framework to leverage target-related textual cues from the description for progressively localizing the target object. The authors use a Large Language Model (LLM) to decompose the input text description into short phrases which are taken as target-related cues and fed into a Conditional Referring module. Besides, Region-aware Shrinking (RaS) loss and Instance-aware Disambiguation (IaD) loss are proposed to facilitate fine-grained cross-modal alignment.

**Strengths:**

1. The motivation is clear and easy to understand. The idea of leveraging target-related textual cues decomposed by LLM for progressive localization makes sense.

2. This work develops multiple consecutive Conditional Referring Modules. RaS and IaD two loss objectives are seamlessly integrated into the framework.

3.  It is novel to implement RaS and IaD two loss objectives by harnessing the capabilities of segmentation foundation models. The SAM[1] does not include strong semantic prior knowledge like GroundingDINO and Grounded-SAM[2], and it is suitable for the weakly-supervised setting.

4. The work shows outstanding object localization ability and outperforms counterparts on three common benchmarks. Especially after refined by the SAM, the results are promising.

[1] Segment anything ICCV2023

[2] Grounding dino: Marrying dino with grounded pre-training for open-set object detection

**Weaknesses:**

1. In table 1, only the results refined by SAM[1] are reported. Considering that FreeSOLO [2] is an unsupervised segmentation model, I am curious about the results refined by FreeSOLO.

2.  This work introduces multiple stages for progressive localization. It may increase the training and inference time of the model.  The time comparison of model training and inference may be needed.

[1] Segment anything ICCV2023

[2] Freesolo: Learning to segment objects without annotations CVPR2022

**Questions:**

1. The mask proposals by SAM are often part of the whole instance.  It will affect the refined results. Does this work do anything special to alleviate the problem? The author can give more details about this issue.

2. Table 1 shows the results of the pseudo labels obtained at the wealy-supervised training stage. Can the authors give the results after pseudo labels based on supervised training?

**Limitations:**

NaN

---

> ### Author Rebuttal · Authors · 2024-08-07
>
> > **[W1]**: In table 1, only the results refined by SAM[16] are reported. Considering that FreeSOLO [40] is an unsupervised segmentation model, I am curious about the results refined by FreeSOLO.
>
> **[Ans]**: Thanks for your advice. Here we present the **mIoU** after applying FreeSOLO refinement, as shown in the table below. The first row displays the results for TRIS[25], while the second row presents the results for our proposed method. Although FreeSOLO's extracted mask proposals are not as accurate as those from SAM, leading to reduced performance, our method still exhibits clear and substantial improvements across three established benchmarks.
>
> | Method     | RefCOCO(val) | RefCOCO+ (val) | RefCOCOg_goolge(val) |
> | :-: | :-:|:-:|:-:|
> | TRIS[25] | 29.7         | 27.2           | 30.5                 |
> | PCNet      | **33.1**     | **30.3**       | **33.8**             |
>
>
>
> > **[W2]**: This work introduces multiple stages for progressive localization. It may increase the training and inference time of the model. The time comparison of model training and inference may be needed.
>
> **[Ans]**: Thanks for your comment. Our method incorporates a multi-stage refinement process, which naturally leads to increased training and inference times. Below, we provide a comparative analysis of the time costs associated with various methods. Notably, the overall time expenditure of our method with three stages remains well within acceptable limits.
>
> | Methods    | Training-Time | Inferring time |
> | :-: |  :-: |  :-: |
> | SAG[14]  | 36.0 h        | 63 ms          |
> | TRIS[25] | 3.0 h         | 35 ms          |
> | PCNet      | 6.0 h         | 42 ms          |
>
>
>
>
>
> > **[Q1]**: The mask proposals by SAM are often part of the whole instance. It will affect the refined results. Does this work do anything special to alleviate the problem? The author can give more details about this issue.
>
> **[Ans]**: You are correct that SAM's mask proposals often represent only a portion of the complete instance. Consequently, even when our method achieves accurate localization, it may not necessarily translate to precise instance masks. To demonstrate this, the last row ("Oracle") of Table 1 presents the mIoU scores achieved by selecting the mask proposal that best aligns with the ground-truth mask. These results support our observation and suggest that integrating a more advanced instance detector, such as GroundSAM [A], could further enhance our method's mask prediction accuracy (mIoU).
>
> > **[Q2]**: Table 1 shows the results of the pseudo labels obtained at the weakly-supervised training stage. Can the authors give the results after pseudo labels based on supervised training?
>
> **[Ans]**: Thanks for your question. We present additional results below. In our main paper, we focused on the quality of pseudo masks generated by our method (i.e., one stage). Here, we further investigate their effectiveness by training the LAVT [44] network using these pseudo masks as supervision. We report the resulting PointM (first row) and mIoU  (second row) metrics on three benchmarks, along with the performance achieved when refining mask predictions with SAM (third row). These findings highlight the potential for further improvement in mask prediction through fully supervised training based on our generated pseudo masks. We will incorporate these results into the revised version of our paper.
>
> | Metric                    | RefCOCO(val) | RefCOCO+ (val) | RefCOCOg_goolge(val) |
> | :-: | :-: | :-: | :-: |
> | **PointM**                | 66.1         | 60.4           | 62.4                 |
> | **mIoU**                  | 44.6         | 40.3           | 41.1                 |
> | **mIoU** (refined by SAM) | **54.5**     | **49.9**       | **51.1**             |
>
> References:
>
> [A] Grounding dino: Marrying dino with grounded pre-training for open-set object detection ECCV2024

---

> > ### Comment · Reviewer_K7SG · 2024-08-08
> >
> > I would like to thank the authors for their detailed responses. All my concerns were thoroughly addressed. Overall, this paper presents a reasonable and well-explained approach to mimicking the progressive process of understanding language instructions. By leveraging visual localization and segmentation tools (e.g., SAM), the method achieves superior language comprehension and target localization, supported by solid experiments and extensive visualization analysis. This work provides clear insights and significant contributions to the field. Therefore, I will maintain my original rating. I also recommend that the authors include suitable explanations and new results in the revision.

---

> > > ### Author Response · Authors · 2024-08-10
> > > **Thanks for your response**
> > >
> > > Thanks very much for your positive feedback and further suggestions! We'll include more explanations and convincing results in our revision as you suggested.

---

### Official Review · Reviewer_Tfcw · 2024-07-02

**Soundness:** 3
**Presentation:** 2
**Contribution:** 2
**Rating:** 5
**Confidence:** 5

**Summary:**

This paper proposes the Progressive Comprehension Network (PCNet) for the WRIS task. this model achieves visual localization by progressively incorporating target-related textual cues for visual-linguistic alignment. Although experimental results have demonstrated the effectiveness of this paper, several aspects of the paper lack clarity.

**Strengths:**

1.	The corresponding experiments validate the effectiveness of the proposed method on three popular benchmarks. the proposed method outperforms existing methods.
2.	The two loss functions introduced in this paper effectively enhance the accuracy of the response maps.

**Weaknesses:**

1.	The Conditional Referring Module (CRM) is implemented through multiple cross-attentions, which is a relatively common approach, lacking novelty.
2.	What is the general idea of classification loss mentioned in the paper? Please explain the fundamental research insights of this work.
3.	In the Region-aware Shrinking (RaS) loss, what does the ⊙ symbol represent in Equation 5? Please clarify its meaning.
4.	This article only conducted ablation experiments on the RefCOCOg dataset, so the ablation experiments are insufficient. It is recommended to validate on multiple datasets (such as RefCOCO+, RefCOCO) to comprehensively evaluate the effectiveness of our method.
5.	The order of the tables is somewhat confusing, with Table 9 preceding Table 6. It is recommended to reorder the tables to make them more in line with logical sequence or reading habits.
6.	Which datasets were the ablation experiments in Tables 7-9 conducted on? Please specify.

**Questions:**

None

**Limitations:**

See the weakness

---

> ### Author Rebuttal · Authors · 2024-08-07
>
> Sincerely thanks for useful comments. To the weaknesses, our response is as follows:
>
> >**[W1]**: Conditional Referring Module (CRM) is implemented through multiple cross-attentions, which is a relatively common approach, lacking novelty.
>
>  Thanks for your comments. We would like to address the concerns about its novelty and highlight its importance in our approach:
>
> - Firstly, the proposed CRM module is well-motivated. In our method, we observe that referring text descriptions typically contain detailed cues on how to localize the target object.  By progressively integrating these  target-related textual cues from the input description, we hope to enhance the target localization step-by-step. Therefore, we introduced the CRM module to model this progressive comprehension process. More importantly, the module is combined with the `RaS` loss to implement instance-level supervision, which has not been explored in previous approaches.
> - The CRM module includes two cross-attentions, each playing a crucial role. The first, vision-to-text attention, aims to obtain vision-attended cue features by incorporating visual context information. The second, text-to-text attention,  modulates the referring query using the target-related textual cue embeddings.  Through the interaction between referring query and different textual cue embeddings, this module is designed to achieve a more discriminative referring embedding. In Table 4, we provide ablation experiments for the module, which also validate the effectiveness of our CRM design.
>
>
> >**[W2]**:  the general idea of classification loss mentioned in the paper.
>
> We sincerely apologize for the less detailed presentation about the Classification loss  $\mathcal{L}\_{\texttt{cls}}$ in **TRIS** [25] due to space restriction. The loss $\mathcal{L}\_{\texttt{cls}}$ aims to establish a global alignment between the visual content and the referring texts by formulating a classification process. We provided a detailed explanation of the idea and implementation of the loss in the **general response**. We will include this in the revision.
>
> >**[W3]**: what does the ⊙ symbol represent in Equation 5.
>
> **Ans**: The $\odot$ denotes the hadamard product. In Equation 5, we multiply the binary mask obtained from SAM and the response map element by element. Thanks for your comments. We will add the explanation in the updated version.
>
> >**[W4]**: this article only conducted ablation experiments on the RefCOCOg dataset, so the ablation experiments are insufficient.
>
> In our main paper, we follow  previous works, e.g., SAG[14] and TRIS[25], to conduct the ablation experiments on the  representative RefCOCOg dataset.  As suggested,  we conducted more ablations on other datasets (i.e., refcoco, refcoco+ datasets)  and shown the results in the following tables. The results on these datasets lead to the same conclusions as those on RefCOCOg, further validating the necessity and effectiveness of each component of our method.
>
> - RefCOCO
>
> |                              |                  |                              |                              |                 |                 |
> | :---------- | :--------- | ----------- | -------- | ------ | ------ |
> | $\mathcal{L}_{\texttt{CLs}}$ | ${\texttt{CRM}}$ | $\mathcal{L}_{\texttt{RaS}}$ | $\mathcal{L}_{\texttt{IaD}}$ | **PointM**      | **mIoU**        |
> | &#10003;                     |                  |                              |                              | 50.3            | 24.6            |
> | &#10003;                     | &#10003;         |                              |                              | 53.5            | 26.4            |
> | &#10003;                     | &#10003;         | &#10003;                     |                              | 56.1            | 28.7            |
> | &#10003;                     | &#10003;         |                              | &#10003;                     | 56.8            | 28.0            |
> | &#10003;                     | &#10003;         | &#10003;                     | &#10003;                     | $\textbf{60.0}$ | $\textbf{31.3}$ |
>
> - RefCOCO+
>
> | $\mathcal{L}_{\texttt{CLs}}$ | ${\texttt{CRM}}$ | $\mathcal{L}_{\texttt{RaS}}$ | $\mathcal{L}_{\texttt{IaD}}$ | PointM          | mIoU            |
> | :--------- | :-------- | ------- | ---------- | -------- | --------- |
> | &#10003;                     |                  |                              |                              | 44.5            | 22.1            |
> | &#10003;                     | &#10003;         |                              |                              | 48.8            | 24.6            |
> | &#10003;                     | &#10003;         | &#10003;                     |                              | 55.3            | 27.1            |
> | &#10003;                     | &#10003;         |                              | &#10003;                     | 52.4            | 25.9            |
> | &#10003;                     | &#10003;         | &#10003;                     | &#10003;                     | $\textbf{58.7}$ | $\textbf{29.2}$ |
>
> >**[W5]**: The order of the tables is somewhat confusing, with Table 9 preceding Table 6.....
>
> Thanks for pointing out the problem. We will reorder the two tables to make them more in line with logical sequence or reading habits in the updated version.
>
> >**[W6]**: Which datasets were the ablation experiments in Tables 7-9 conducted on? Please specify.
>
> We conduct the ablation experiments on **Refcocog** (google split) dataset as described in `lines 275-276`.

---

> > ### Comment · Reviewer_Tfcw · 2024-08-12
> >
> > Thanks for the rebuttal. My concerns are addressed, therefore I increase my score.

---

> > > ### Author Response · Authors · 2024-08-12
> > >
> > > Thanks very much for your positive feedback! We deeply appreciate that you decided to increase the score. We'll include the explanations and more convincing results in our revision according to your careful comments.

---

### Official Review · Reviewer_kTSb · 2024-07-12

**Soundness:** 2
**Presentation:** 1
**Contribution:** 2
**Rating:** 5
**Confidence:** 3

**Summary:**

Inspired by human's step-by-step cognitive process for localizing a target object in an image, the paper proposes Progressive Comprehension Network (PCNet) for the task of weakly-supervised referring image segmentation (WRIS) where text is the only supervision signal. PCNet first decomposes a long, complex text description into multiple target-related cues using a large language model (LLM), and the specialized designed module, Conditional Referring Module (CRM), progressively refines the text-to-image response map using the generated text cues in multiple stages. Region-aware Shrinking (RaS) is introduced for constraining the latter stage to have a more compact, target-related response map, and Instance-aware Disambiguation (IaD) loss is proposed for differentiating between the target object and the other objects in the input image. The experiment results show the superiority of the proposed method.

**Strengths:**

- The paper provides extensive experiment results proving the validity of the proposed method, PCNet, and also ablation studies showing the contributions of main components of the model.
- Figures help readers understand the proposed method.

**Weaknesses:**

- The proposed method uses the contrastive loss of TRIS, but not generates independent response maps from a positive cue and L negative cues. In Equation (3) and (4), it integrates all the text cue features and generates a single response map R_n that contains information from both positive and negative cues. This does not make sense. How did the authors differentiate between positive and negative cues when computing the classification loss L_Cls? (This is why I rated "poor" for presentation).
- IaD, the loss to differentiate between the target object and other objects in the same image is similar to the calibration loss used in TRIS; the idea is the same while the calibration loss uses CLIP image-text similarity for differentiation. The authors should have discussed about this loss in the paper, and also done an ablation study comparing the two losses.
- The PCNet uses the mask proposals generated by a pre-defined segmentation mask generator such as FreeSOLO and SAM. We cannot ignore the possibility of knowledge being distilled from the mask generator into PCNet, especially through RaS and IaD losses, main contributions of the paper. More specifically, we cannot be sure that the performance gains from using RaS and IaD do not come from the mask generator's knowledge.

**Questions:**

Please rebut the above weaknesses. I also have minor questions:
- In Figure 2, the arrow comes from Q_{n+1} to Cls. How is Q_{n+t} used for computing the classification loss?
- How did the authors sample L negative textual cues? one cue from each of the examples in the same mini-batch?
- I am curious about the rationale behind Equations (2) and (3). Why did the authors use residual connection only after the mlp layer, not also after the self-attention layer like in Transformer?
- Does the order in which text cues are used affect the performance of the model? E.g., is there a performance difference between feeding into the CRM module in the order of q_0, q_1, q_2 and in the order of q_0, q_2, q_1?

**Limitations:**

The paper addressed the limitations of the work.

---

> ### Author Rebuttal · Authors · 2024-08-07
>
> Sincerely thanks for your useful comments. To the weaknesses and questions, our response is as follows:
>
> **[W1]**:
>
> Thank you for you careful comment. We sincerely apologize for the less detailed presentation regarding the  loss  $\mathcal{L}_{\texttt{cls}}$ in **TRIS** [25] due to space restrictions. We provided a detailed  explanation of the idea and implementation of this loss in the **general response**.  We will include this loss explanation in the revision.
>
> **[W2]**:
>
> The idea of two loss functions is different and there are  two essential differences between them:
>
> - **Motivation**. The calibration loss in TRIS is used to suppress noisy background activations and thus help to re-calibrate the target response map.  In contrast, in our method, we observe that, multiple referring texts corresponding to different instances in one image may locate the same (or we say overlapping) instances (described in `lines 210-212`), due to the lack of instance-level supervision in WRIS.
>
> - **Implementation**. The calibration loss adopts the global CLIP score of image-text to implement a simple contrastive learning for revising the response map. Differently, we simultaneously infer the response maps of different referring texts from the same image, and obtain the **instance-level** localization results by choosing the  mask proposal with max alignment score. `IaD` can achieves better localization accuracy with instance-level differentiation.
>
>
> - To further verify the superiority of our loss, we conduct **two-groups ablations** on the **RefCOCO (val)** dataset. The first ablation used `PCNet` without `IaD` loss as the baseline, and  the second used the TRIS without calibration loss. We then separately introduced these two loss functions  for comparison. Both ablations demonstrate  that the `IaD` loss not only refines the response map (mIoU metric) but also significantly improves the localization accuray (PointM metric).
>
>   - baseline  (CLS+CRM+RaS)
>
>     | Baseline | Calibration loss | IaD loss | PointM   | mIoU  |
>     | - | :-:|:-:|:-:|:-:|
>     | &#10003; |  |  | 56.1 | 28.7   |
>     | &#10003; | &#10003;  |  | 56.9     | 29.9   |
>     | &#10003; |  | &#10003; | 60.0 | 31.3 |
>
>   - baseline (CLS)
>
>     | Baseline | Calibration loss | IaD loss | PointM | mIoU |
>     | - | :-:|:-:|:-:|:-:|
>     | &#10003; |   |  | 50.3   | 24.6 |
>     | &#10003; | &#10003;  |  | 51.4 | 26.4 |
>     | &#10003; |  | &#10003; | 54.7 | 26.3 |
>
> **[W3]**:
>
> We would address your concerns from the following four aspects:
>
> - Mask generator does not contain any semantic information. Though it can be used as post-processing method to improve the completeness of the mask, here we aims to improve our localization accuracy by introducing suitable constraint based on the mask proposals.
>
> - Our core idea is to improve the understanding of textual cue information by progressive process. The ablation experiments (Table 2) demonstrate that even without **RaS** and **IaD** loss, our approach improves localization to some extent. However, relying only on global classification loss is suboptimal due to its lack of modeling at the instance level.  Therefore, we introduce **RaS**, which leverages mask proposals to model multi-stage refinement and enhance instance-level cross-modal alignment.
>
> - In lines `200-204`, we point out that a complete mask proposal can improve the mask quality to some extent, but it is not the core of our `RaS` loss.  In the following, we conducted a quantitative analysis on **RefCOCO** val set.
>
>   - The first line denotes the results of baseline (include classification loss and CRM module). In the second line, we utilize the mask proposal generated by SAM (refer to `line-189`) as pseudo mask to calculate  the `IoU` loss between it and the response map. The third line denotes the results of our `RaS` loss.
>   - While the guidance of the mask proposal can improve the accuracy of the response map to some extent, there is a significant gap with the `RaS` loss on the localization performance. Besides, as shown in Table 1, even with the less accurate mask proposal by FreeSOLO, our methods still achieve superior localization performance. These results show that the performance improvement of our method is not exactly due to the introduction of mask generator.
>
>   | Methods    | PointM   | mIoU   |
>   | - | :-:|:-:|
>   | Baseline (CLs +CRM)  | 53.5   | 26.4   |
>   | Baseline  + IoU_loss | 54.5   | 27.9   |
>   | Baseline  + RaS      | 56.1 | 28.7 |
>
> - In addition, the `IaD` loss only utilizes the mask proposal as a localization result to derive the loss formula, there is no exploitation of mask knowledge.
>
> **[Q1]:**
>
> I guess the question concerns how $Q_{n+1}$, not $Q_{n+t}$ (absent in Fig. 2), is used for calculating the Classification loss. As explained in the **general response**, $Q_{n+1}$ in stage $n$ contains one positive and $L$ negative referring embeddings per image. Given this known text-image correspondence, computing the cross-entropy loss is straightforward.
>
> **[Q2]:**
>
> Leveraging the **refer_id** and **image_id** annotations in datasets, which link texts to unique instances, we randomly sample texts referring to different instances from the original. Simultaneously the cues for each text in the batch is sampled (one cue is used for referring embedding modulation at each stage).
>
> **[Q3]**:
>
> About the residual design, we draw inspiration from the previous works (like TRIS[25] and DenseCLIP[A]). The design can integrate useful cross-modal information while preserving the original modal information from degradation.
>
> **[Q4]**:
>
> The order of the text inputs would not distinctly affect the performance of our method. In the table below, we conducted a quantitative comparison on the **RefCOCO** (val) to verify this.
>
> | Order      | PointM | mIoU |
> | :-: |:-: | :-: |
> | q1, q2, q3 | 60.0   | 31.3 |
> | q1, q3, q2 | 59.7   | 31.2 |
>
> References:
>
> [A] Denseclip: Language-guided dense prediction with context-aware prompting CVPR2022

---

> ### Comment · Reviewer_kTSb · 2024-08-10
>
> Thank the authors for the detailed response. The original formulation of the classification loss was quite confusing, but now it makes sense. Also, the comparison with the baseline + IoU loss convinces me of the contribution of the RaS loss. The authors also addressed my other concerns. It depends on further discussion with other reviewers, but I am leaning towards raising my rating. Please add the conducted comparison experiments with the calibration loss in the final draft.

---

> > ### Author Response · Authors · 2024-08-11
> > **Thanks for your response**
> >
> > Thanks very much for your positive feedback that we have addressed your concerns through our response. Your decision to upgrade our score is deeply appreciated. Please let us know if you have further questions. We'll also include more explanations and new results in our revision as you suggested.

---

> ### Author Response · Authors · 2024-08-13
>
> Thanks for your response and your willingness to consider raising the score. We genuinely appreciate your time and the valuable feedback. We are pleased that our rebuttal has addressed your concerns. However, we notice that your current rating still suggests that our work  is still not qualified to be accepted. Considering that the discussion period is ending soon, if there are any remaining issues or concerns, we would greatly appreciate the opportunity to discuss them further with you.

---

### Official Review · Reviewer_oLx4 · 2024-07-14

**Soundness:** 2
**Presentation:** 2
**Contribution:** 2
**Rating:** 7
**Confidence:** 4

**Summary:**

This paper proposes a Progressive Comprehension Network for weakly-supervised referring image segmentation, which mimics the human process of progressive understanding by breaking down sentences into segments and gradually narrowing down the target range. The main contributions include: A multi-stage Conditional Referring Module to progressively comprehend text cues. A Region-aware Shrinking Loss to constrain the target region to gradually shrink. An Instance-aware Disambiguation Loss to eliminate overlap between different instances.

**Strengths:**

1. Presentation: The majority of the paper is well-written and easy to follow, providing clear explanations of the key concepts.
2. Motivation: The motivation is reasonable and convincing. By mimicking the human process of understanding concepts from coarse to fine, the method progressively refines the final mask. Directly requiring the mask to continuously shrink could lead to trivial solutions, and two losses are introduced to further support this pipeline. The proposed method aligns well with the motivation.
3. Experiments: Comprehensive ablation experiments demonstrate the effectiveness of the proposed modules

**Weaknesses:**

1. The concept of mimicking the human process of understanding concepts from coarse to fine-grained, along with multi-stage refinement for different parts of sentences, has been explored in prior REC research [1]. This approach involves parsing sentences into multiple parts and conducting multi-stage refinement. Applying REC's ideas [1] to WRIS can also be considered a contribution, but it should be thoroughly discussed in comparison to [1], highlighting any differences, and cited appropriately.
2. There is a lack of discussion and citation of recent WRIS work, such as [2], in related work and the main text. It is necessary to discuss and cite the latest WRIS work in both the related work section and the main table.

3. Section 3.4 is not entirely understandable. I suggest the authors re-describe this section to make it clearer and easier to understand.

4. The top of Fig.2 is not clear enough, especially the subscripts a and d, I suggest the author re-design this part.


 [1] Yang S, Li G, Yu Y. Dynamic graph attention for referring expression comprehension[C]//Proceedings of the IEEE/CVF International Conference on Computer Vision. 2019: 4644-4653.

 [2] Dai Q, Yang S. Curriculum Point Prompting for Weakly-Supervised Referring Image Segmentation[C]//Proceedings of the IEEE/CVF Conference on Computer Vision and Pattern Recognition. 2024: 13711-13722.

**Questions:**

1. Why is the range of Equation 6 set to [0,1]? Can the combination of other proposals (denoted as 𝑚_𝑏 ) be simply approximated by the complement of the foreground, since the proposals segmented by SAM nearly cover the entire image? In this case, if IoU(𝑅_𝑛,𝑚_𝑓) < IoU(𝑅_𝑛,𝑚_𝑏), wouldn't Equation 6 be greater than 1?
2. Does 𝑚_𝑓 change at different stages in Equation 7, for instance, when different proposals are selected at different stages? What issues could this cause? In Equation 7, are 𝑛 and 𝑛+1 reversed? According to Section 3.3, the ambiguity score should be lower at later stages.
3. Section 3.1 mentions that there are five phrases for each sentence, and Section 3.2 states that each stage uses one text cue, so there should be five stages. Why do the implementation details mention only three stages?
4. In Section 3.4, how do you sample extra 𝑁_𝑑 texts? Are they randomly sampled from multiple text descriptions corresponding to the same image? How do you ensure that the sampled text descriptions correspond to different target referents? Because in refcoco series dataset, one referent may have multiple text expressions.
5. In the bottom part of Tab.1, are the numbers all obtained by using a single peak point as the prompt for SAM? Did you try using the response map as the prompt for SAM?

**Limitations:**

Yes.

---

> ### Author Rebuttal · Authors · 2024-08-07
>
> Sincerely thanks for useful comments. To the weaknesses and questions, our response is as follows:
>
> **[W1]**:
>
> Thank you for your advice. Although both DGA [A] and our method adopt multi-stage refinement, there are significant differences:
> - The motivation is different. DGA focuses on the fully-supervised visual grounding task and aims to model visual reasoning on top of the relationships among the objects in the image. In contrast, our work addresses the weakly RIS task and aims to alleviate the localization ambiguity by progressively integrating fine-grained attribute cues.
> - The implementation is different. In the absence of mask, we build the relation between the response map of different stages using the proposed  RaS loss.
> - We will discuss and include their differences in the revision.
>
> **[W2]**:
>
> Thank you for your advice. PPT [B], appearing on Arxiv near the NeurIPS deadline, is considered contemporaneous work. It introduces a parameter-efficient point prompting framework. The Table below presents a comparison on the val sets using **mIoU**. While our method utilizes down-sampled SAM mask proposals, we also tested with full-resolution masks for fairness, aligning with PPT. In both scenarios, our method demonstrates superior target localization compared to PPT. We will discuss  and include their differences  in the revision.
>
> | Methods                      | val (RefCOCO) | val (RefCOCO+) | val (RefCOCOg_Google) |
> | -|:-:|:-:|:-:|
> | PPT                          | 46.7          | 45.3           | 42.9                  |
> | Ours (down-sampled SAM mask) | **48.6**      | 43.3           | **45.4**              |
> | Ours (raw SAM mask)          | **52.2**      | **47.9**       | **48.3**              |
>
> **[W3 & W4]**:
> - Explanation about `Sec. 3.4`
>   - **Motivation**. Considering that each image generally contains multiple instances, each paired with its reference description, the weak discriminative capability of the model in WRIS often leads to multiple texts for different instances in a single image activating the same region (i.e, response map overlap in Line-209). Thus, we propose the **IaD** loss in `Sec. 3.4`  to encourage referring texts for different instances (in the same image) to activate different regions.
>   - **Implementation**. Since this loss function is applied individually to each image sample, for clarity, we omitted the batch concept and consider only one sample, and omitted the subscript `a` used in the Line 215-Line 227. Specifically, given one image-text pair $\\{\mathbf{I}, \mathbf{T}\\}$, we sample $N\_d$ (By default,  $N\_d=1$)  texts for current image sample, which means that we have two referring texts (i.e., $\mathbf{T}$ and $\mathbf{T}\_{d}$ )  for the image $\mathbf{I}$. Then, we can obtain response maps $\mathbf{R}$ and $\mathbf{R}\_{d}$ according to the description in `Sec 3.2`, and get the alignment scores $S$ and $S\_{d}$ according to `Eq. (5)`. Afterwards, we get the instance indexes $\texttt{argmax}(S)$ and $\texttt{argmax}(S\_{d})$ as the proposal predictions by them.  Considering $\texttt{argmax}(\cdot)$ is a non-differentiable operation during gradient backward, we adopt a differentiable implementation by `Eq. (8)`. Finally, we can impose MSE loss on the indexes to encourage the two texts (in the same image) to activate different regions by `Eq (9)`.
> - We will revise the figure to make the subscripts clear in the revision.
>
> **[Q1]**:
>
> - Thanks for pointing out this issue. This is a typo, and the range shold be [0,2], as in some cases, it may occur that $\text{IoU}(𝑅_𝑛,𝑚_𝑓) < \text{IoU}(𝑅_𝑛,𝑚_𝑏)$.
>
> - It is less feasible. The shrinking loss constraint the response map toward the compact orientation while maintaining located in the foreground area.  $𝑚_𝑏$ include less foreground part and more background than the $m_f$. It would lead to more background activation when choosing the $𝑚_𝑏$ as the complement of the foreground.
>
> **[Q2]**:
>
> It is possible to have different instances for different stages. According to our statistical result, this issue occurs in a small number of cases (less than 10%) and has minimal impact on loss optimization. We will include this analysis  and revise the Eq. (7) in the revision.
>
> **[Q3]**:
>
> LLM outputs for referring texts often vary in attribute cue count (2-5). To enable parallel training, we standardize them to 5 cues via repetition padding (`Sec. 3.1`). The Tab. 3's ablation study on stage numbers shows optimal performance with 3 stages, thus our framework's implementation utilizes three stages.
>
> **[Q4]**:
>
> In addition to the standard training batch ($B$ image-text pairs), we sample $N_d$ extra texts ($N_d=1$) per image. Leveraging the **refer_id** and **image_id** annotations in refcoco datasets, which link texts to unique instances, we randomly sample texts referring to different instances from the original.
>
> **[Q5]**:
>
> All results utilize a single peak point as the SAM prompt. As suggested, we compared different prompts on the validation set (see table below). Using the response map as the prompt proved sensitive to thresholds, impacting noise ratios and yielding inferior performance compared to the single peak point prompt.
>
> | Methods                      | val (RefCOCO) | val (RefCOCO+) | val (RefCOCOg_Google) |
> | -| :-:|:-:|:-:|
> | single peak + SAM            | **52.2**      | **47.9**       | **48.3**              |
> | pseudo mask (thre=0.8) + SAM | 37.7          | 33.4           | 38.4                  |
> | pseudo mask (thre=0.5) + SAM | 41.1          | 35.6           | 39.3                  |
> | pseudo mask (thre=0.2) + SAM | 39.4          | 35.3           | 35.6                  |
>
> **References**
>
> [A] Dynamic graph attention for referring expression comprehension--ICCV 2019
>
> [B] Curriculum Point Prompting for Weakly-Supervised Referring Image Segmentation--CVPR. 2024

---

> > ### Comment · Reviewer_oLx4 · 2024-08-13
> >
> > Thank you for your detailed response, which addressed most of my concerns. I would like to further discuss the IaD loss and Q4. In GroupViT, a similar operation is used to backpropagate gradients, but they do this to achieve hard assignment during inference, while the gradients are computed in a soft form. However, in this paper, there is no need for hard assignments during inference. Is there a simpler equivalent form to replace the current IaD loss?
> >
> > Regarding the answer to Q4, can I understand that you utilize the prior of different texts corresponding to different instances within the same image using **refer_id** ? Is it used in TRIS?
> >
> > I will raise my score if you can provide an explanation. Thank you very much!

---

> ### Author Response · Authors · 2024-08-13
>
> Thanks very much for your positive feedback that we have addressed your most concerns and your willingness to raise the score. Here we give further explanations about IaD loss and Q4.
>
> - IaD loss. Thanks for your careful comments!  In GroupViT, the Grouping Block groups similar semantic regions using the hard assignment strategy during the group tokens training and inferring process. In our IaD loss, we also adopt a similar hard assignment for deriving the loss function.  The reason is that we aims to get the pseudo mask prediction by the accurate peak value point (i.e., the hard assignment results) instead of relying on whole score distribution $S(\cdot)$ (e.g., $S_{a}$, $S_{d}$ in Section 3.4). Thus utilizing the hard assignment to derive the IaD loss well matches our purpose, which helps rectify the ambiguous localization results. If we use the soft assignment (e.g., measuring KL divergence between  $S_{a}$ and  $S_{d}$), though the equivalent may be simpler, it not only does not match our purpose but also introduces more tricky components for optimization (e.g., extra distribution regularization is required). In order to verify the argument, we conduct a comparison on RefCOCOg(G) val dataset as follows. The KL_Loss even causes a slight decline, while the IaD loss brings a clear improvement on localization.
>
>   | Methods                                 | PointM | mIoU |
>   | :-------------------------------------- | ------ | ---- |
>   | $\mathcal{L}_{\texttt{CLs}}$            | 51.7   | 25.3 |
>   | $\mathcal{L}_{\texttt{CLs}}$ + KL_loss  | 51.2   | 24.8 |
>   | $\mathcal{L}_{\texttt{CLs}}$ + IaD_loss | 53.1   | 26.6 |
>
> - Q4. Yes, you are right. `refer_id` is important information for the data_loader sampler and adopted in TRIS and SAG. Here, we further consider and utilize the prior (i.e., different texts corresponding to different instances within the same image should activate different regions) for localization optimization.
>
>  If there are any additional clarifications needed, please do not hesitate to reach out.

---

> > ### Comment · Reviewer_oLx4 · 2024-08-13
> >
> > Thank you for your detailed response, which addressed all my concerns. I decide to raise the score. Please include the previously mentioned discussions in the final revision.

---

> > > ### Author Response · Authors · 2024-08-13
> > > **Official Comment by Authors**
> > >
> > > We thank Reviewer oLx4 for reviewing our work and for raising the review score. We really appreciate it. We will include the discussions in our revision.

---

### Author Rebuttal · Authors · 2024-08-07

**To Reviewers and AC:**

We extend our sincere gratitude to all the reviewers (**R1**-oLx4, **R2**-kTSb, **R3**-Tfcw, and **R4**-K7SG) for your time and insightful reviews, which help us emphasize the contributions of our work and revise the presentation. We are encouraged to hear that the reviewers found the work is well-motivated with good presentation and contribution (**R1**, **R4**), as well as the comprehensive experimental evaluation and commendable performance (**R1**, **R2**, **R3**, **R4**). We have methodically addressed each point in our individual responses, hoping that we can address your concerns.

Here we first address broader questions:

**Explanation about the idea and implementation of classification loss $\mathcal{L}\_{\texttt{cls}}$ in TRIS** [25]:

- Our work consists of multiple stages and utilizes $\mathcal{L}\_{\texttt{cls}}$ in TRIS at each stage independently for response maps optimization. Here, we omit the index of stage n for clarity.

- $\mathcal{L}\_{\texttt{cls}}$ formulates the target localization problem as a classification process to differentiate between positive and negative text expressions. While the referring text expressions for an image are used as positive expressions, the referring text expressions from other images can be used as negative expressions for this image.  Thus, given a batch (i.e., B) of image samples , each image sample is mutually associated with one positive reference text (i.e., a text describing a specific object in the current image) and mutually exclusive with $L$ negative reference texts (texts that are not related to the target object in the image). Note that the number of batches is equal to the sum of the positive samples and the negative samples (i.e., $B = 1 + L$).

- Speficially, in each training batch,  $B$ (i.e., $1+L$) image-text pairs $\\{ \mathbf{I}\_i, \mathbf{T}\_i \\}\_{i=1}^{B}$ are sampled. Through the language and vision encoders, we can get  referring embeddings ${\mathbf{Q}} \in \mathbb{R}^{B \times C}$ and image embeddings ${\mathbf{V}} \in \mathbb{R}^{B \times H \times W \times C}$. Then, we obtain the response maps ${\mathbf{R}} \in \mathbb{R}^{B \times B \times H \times W}$  by applying similarity calculation and normalization operation. After the pooling operation as done in TRIS, we further obtain the alignment score matrix ${\mathbf{y}} \in {\mathbb{R}}^{B \times B}$.  According to the $\mathcal{L}\_{\texttt{cls}}$, for $i\_{th}$ image in the batch, there is a prediction score $\mathbf{y}{[i, :]}$, where $\mathbf{y}{[i, i]}$ predicted by the corresponding text deserves a higher value (i.e, **1** positive one) and the others deserve lower values (**L** negative ones).

  Then `Classification` loss for the $i\_{th}$ image from the batch can be formulated as cross-entropy loss:
 $$
\mathcal{L}\_{\texttt{cls}, i} = - \frac{1}{B} \sum\_{j=1}^{B}\left( \mathbb{1}\_{i=j} \log\left( \frac{1}{1+e^{-\mathbf{y}{[i, j]}}} \right) + (1-\mathbb{1}\_{i=j}) \log \left(\frac{e^{-\mathbf{y}{[i, j]}}}{1+e^{-\mathbf{y}{[i, j]}}} \right) \right),
 $$
​        and the `Classification` loss for the batch can be formulated as:
$$
\mathcal{L}\_{\texttt{cls}}= \frac{1}{B} \sum\_{i=1}^{B} \mathcal{L}\_{\texttt{cls}, i}
$$
​        The $i$ denotes the index for the visual image and the $j$ denotes the index for the text.



We address the raised concerns below and will revise our paper according to all comments. Please let us know if you have further questions.

Regards,

Authors (paper 891)

---

### Decision · Program_Chairs · 2024-09-25

**Decision:**

Accept (poster)

**Comment:**

Initially, the paper received mixed reviews, but following the author's rebuttal, the reviewers have reached a positive consensus. Each reviewer noted that the authors effectively addressed all concerns raised in their initial feedback. Two reviewers now strongly support the paper with a score of 7. The paper's contributions are recognized for their clear motivation, strong presentation, originality, and extensive experimental support. The AC also agrees that the paper exceeds the acceptance standards.